# Acute leucocyte, muscle damage, and stress marker responses to high-intensity functional training

João Henrique Gomes[1]*, Renata Rebello Mendes[2◉], Crystianne Santana Franca[2], Marzo Edir Da Silva-Grigoletto[3], Danilo Rodrigues Pereira da Silva[3], Angelo Roberto Antoniolli[1,4], Ana Mara de Oliveira e Silva[1,2◉], Lucindo José Quintans-Júnior[1,4◉]

1 Health Sciences Graduate Program, Federal University of Sergipe, Aracaju, Sergipe, Brazil, 2 Nutrition Department, Federal University of Sergipe, São Cristóvão, Sergipe, Brazil, 3 Department of Physical Education, Federal University of Sergipe, São Cristóvão, Sergipe, Brazil, 4 Laboratory of Neuroscience and Pharmacological Assays, Federal University of Sergipe, São Cristóvão, Sergipe, Brazil

◉ These authors contributed equally to this work.
* profjhgomes@gmail.com

**Data Availability Statement:** All relevant data are within the manuscript and its Supporting Information files.

## Abstract

### Background

High-intensity functional training (HIFT) has become more popular, and the number of practitioners has increased; however, it remains unclear whether perturbations in the immune parameters occur, even after one single bout. Our aim was to examine acute leucocyte, muscle damage, and stress marker responses following a single 'Cindy' workout session, and compare the results between novice and experienced participants.

### Material and methods

Twenty-three HIFT practitioners (age 31.0 ± 1.0 years) completed the 'Cindy' workout. They were categorized as novice (3–8 months of experience; n = 10) and experienced ($\geq$18 months; n = 13). White blood cell (WBC) count, plasma creatine kinase (CK) activity, blood cortisol level, and lactate concentration were measured. Blood analysis was performed before (pre-ex), immediately after (post-ex), 30 min after (post-30 min), and 24 h after (post-24 h) a single 'Cindy' workout session.

### Results

WBC count was higher post-ex (6.8 to $11.8 \times 10^3/\mu L$) and returned to baseline values within post-30 min (p<0.01). Neutrophil (3.3 to $4.5 \times 10^3/\mu L$) and lymphocyte levels (2.8 to $5.9 \times 10^3/\mu L$) were higher post-ex and returned to baseline values after post-24 h, yet lymphocytopoenia ($2.2 \times 10^3/\mu L$) was observed at post-30 min (p<0.01). CK increased post-ex (174.9 to 226.7 $U.L^{-1}$) and remained elevated post-24 h. Cortisol (14.7 to 17.0 µg/dL) and lactate (1.9 to 13.5 $mmol.l^{-1}$) responses increased post-ex, but only the lactate level was reduced at post-30 min (p<0.01). The experienced participants had higher WBC, lymphocyte, and cortisol concentrations post-ex than the novice ones (p<0.01).

**Funding:** This study was financed in part by the Coordenação de Aperfeiçoamento de Pessoal de Nível Superior – Brazil (CAPES – Finance Code 001). The funders had no role in study design, data collection and analysis, decision to publish, or preparation of the manuscript.

**Competing interests:** The authors have declared that no competing interests exist.

## Conclusions

A single HIFT session elicited significant acute perturbations in WBC count, stress markers, and muscle tissue, which is like other similar regimens. Importantly, the experienced participants showed greater lymphocyte and cortisol responses than the novice ones.

## Introduction

High-intensity functional training (HIFT) is a form of exercise enjoyed recreationally by participants of varying levels of fitness, training experience, age, and lifestyles, and it is also considered a sport on its own [1–3]. This exercise regimen characterized by high intensity, constant variation, and functional movement is often performed in rapid, successive repetition with limited or no recovery time [4–7]. HIFT is based on the concept of increased work capacity over time while using a variety of exercise modalities, including mono-structural (e.g. running, rowing, etc.), as well as body weight movements (e.g. squats, push-ups, etc.) and weightlifting derivatives (e.g. snatch, shoulder press, deadlift, etc.) [4, 6, 8].

HIFT mixes endurance, power, and strength within the same sport and exercise, at the same time, making it unlike any other. It may be the most comprehensive sport modality, as it requires all domains of physical conditioning [9]. HIFT programs are designed to improve parameters of general physical fitness (e.g. cardiovascular endurance, strength, body composition, flexibility, etc.) and performance (e.g. agility, speed, power, strength, etc.) [8, 10]. Nevertheless, some features of HIFT programs do not seem to follow the norms for safe and proper muscular fitness development, which leads to high health risks and an elevated risk of injury [7, 11–13].

The most popular among these HIFT regimens is CrossFit®, a short-duration, high-volume and high-intensity exercise program [3, 6]. Its training is organized as daily sessions called 'workouts of the day' (WODs) which usually take 20 min or less, and it is done with short or no rest periods, causing the participants to be susceptible to changes in immune parameters, endocrine activities, and redox status [5, 9, 14, 15]. The workouts in this program are based on three modalities: gymnastics, metabolic conditioning, and weightlifting [16].

Recently, investigators have demonstrated that a HIFT bout elicited significant increases in interleukin-6 (IL-6), IL-10, lactate, and glucose concentrations after WODs [5]. An acute blood oxidative stress response comparable to a traditional bout of high-intensity treadmill running in male HIFT practitioners was also observed [14]. Similarly, Heavens et al. [17] showed that a high-intensity, short-rest protocol (with characteristics similar to those of a HIFT session) elicits significant increases in indirect blood markers of muscle damage. As far as we know, only one study has chronically evaluated immunological responses in HIFT practitioners [13]. Despite the popularity of HIFT, very few studies have taken the impact of a single training session on physiological variables, such as metabolic markers, immune biomarkers, stress markers, redox status, and hormonal aspects into consideration [2, 5, 9, 14].

Therefore, we focused on outcomes, like immune parameters, during a 24-h recovery period. To the best of our knowledge, this is the first study to analyze the effects of a single HIFT session on parameters related to immune cell changes, such as white blood cell (WBC) count. Thus, the present study aimed (1) to examine the WBC count, plasma creatine kinase (CK) activity, cortisol concentration, and lactate response following a single 'Cindy' workout session, and (2) to compare the differences between novice and experienced participants. Specifically, the hypotheses tested were as follows: a single 'Cindy' workout session would induce

perturbations in immune parameters after a training session, and novice participants would be more susceptible to these changes when compared to experienced participants.

## Materials and methods

### Experimental approach to the problem

This study was conducted over a total of five visits, comprehending anthropometric evaluation, familiarization with the WODs, and food record explanation (visit 1); one repetition maximum (1RM) test (visit 2); fitness test (visit 3); 'Cindy' workout (visit 4); and a blood sample collection 24h later (visit 5). Prior to and after the training session, blood samples were collected from the antecubital vein and transferred to vacuum tubes (Vacutainer; Becton Dickinson, USA) for further analysis of immune parameters, endocrine activities, and biochemical responses.

On the first visit, volunteers underwent anthropometric measurements (body mass, body height, and skinfold thicknesses) [18], familiarization with the 'Cindy' workout, and eating pattern assessment. For the food record, participants were asked to record three non-consecutive days of their food intake so that their energy and macronutrient intakes could be calculated later using the software "Dietpro" version 5.1. All participants were instructed on how to properly log their food, snacks, and drinks. They received illustrative material with explanations about portion sizes and dietary technical, and they were told not to consume caffeine-rich drinks or foods (e.g. tea, coffee, and chocolate), or alcoholic beverages for 24 h in order to avoid any interference with body hydration. On the second visit (after 48 h), the participants performed the deadlift exercise for 1RM test [19]. After a 24-h rest, on the third visit, the participants performed the yo-yo intermittent recovery test (yo-yo IR1) [20], during which they were monitored through a heart rate (HR) monitor (Polar Team Pro, Kempele, Finland), and the maximum HR was recorded. Strong verbal encouragement was provided to elicit maximal effort during both tests. All tests took place at the same CrossFit® gym. On visit 4, each test was carried out between 6h00 AM and 8h00 AM to avoid circadian cycle influences. Volunteers were asked to refrain from physical exercise for 48 h. Females were tested in the early follicular phase of the menstrual cycle (2–7 days after the onset of menses) to minimize the effect of hormonal fluctuations on outcome measures.

### Subjects

Healthy volunteers were recruited from a CrossFit® affiliate gym ('CrossFit® Quest', Brazil). Inclusion criteria for participant selection were at least 3 months of experience in HIFT routines and the ability to perform the 'Cindy' workout. Novice (NOV) participants were defined as those with 3–8 months of experience, while experienced (EXP) participants were those who had more than 18 months of experience. These timelines were selected to ensure a significant gap in experience level between the two groups of participants [21]. Subjects were excluded if they presented (a) 9–17 months of experience; (b) any injury or motor impairment; (c) any cardiovascular, metabolic, or neurological diseases; (d) the use of any type of medication or performance-enhancing drugs; (e) the use of supplements containing antioxidant compounds in the last six weeks, as well as those considered ergogenic [22], such as caffeine, creatine, beta-alanine, nitrate, and bicarbonates in the last four months; (f) incomplete 'Cindy' workout on visit 4; (g) onset of symptoms of upper respiratory tract infections at any time during the experimental design; and (h) failure to consume the pre-workout standardized breakfast on visit 4.

Ethical approval was obtained from the local review board (Research Ethics Committee of Federal University of Sergipe, process n°. 3.087.955/2018), and the complete project was

**Table 1. Characteristics of the participants (M ± SEM [SD]).**

| Variables | EXP (13) | NOV (10) | p-value | All (23) |
|---|---|---|---|---|
| Age (years) | 31.1 ± 1.4 [4.9] | 30.9 ± 1.5 [4.8] | 0.932 | 31.0 ± 1.0 [4.8] |
| Sex | 7 M 6 F | 5 M 5 F | 0.855 | 12 M 11 F |
| Experience (months) | 28.5 ± 1.8 [6.4] | 6.0 ± 0.5 [1.5] | 0.001 | 18.7 ± 2.5 [12.4] |
| Weight (kg) | 70.8 ± 3.7 [13.3] | 78.5 ± 4.5 [14.1] | 0.197 | 74.2 ± 2.9 [13.9] |
| Height (cm) | 1.69 ± 0.03 [0.10] | 1.69 ± 0.03 [0.09] | 0.908 | 1.69 ± 0.02 [0.09] |
| BMI (kg/m$^2$) | 24.4 ± 0.7 [2.5] | 27.3 ± 0.8 [2.7] | 0.017 | 25.7 ± 0.6 [2.9] |
| Body Fat (%) | 17.4 ± 1.4 [5.1] | 22.5 ± 1.7 [5.4] | 0.031 | 19.6 ± 1.2 [5.7] |
| Deadlift (kg) | 131.6 ± 11.5 [41.6] | 105.2 ± 8.7 [27.6] | 0.098 | 120.6 ± 7.9 [37.9] |
| Yoyo RL1 (m) | 507.7 ± 57.9 [208.7] | 336.0 ± 52.4 [165.7] | 0.045 | 424.3 ± 43.0 [206.3] |
| VO$_{2max}$ (ml/kg/min) | 40.7 ± 0.5 [1.8] | 39.2 ± 0.4 [1.4] | 0.045 | 40.0 ± 0.4 [1.7] |
| HR$_{max}$ (bpm) | 183.2 ± 2.9 [10.3] | 189.4 ± 2.1 [6.6] | 0.111 | 185.9 ± 1.7 [9.3] |

M = mean; SEM = standard error of the mean; SD = standard deviation.

M = male; F = female; BMI = body mass index; Yoyo RL1 = yoyo recovery test level 1; VO$_{2max}$ = maximum oxygen consumption; HR$_{max}$ = heart rate maximum;

EXP = experienced group; NOV = novice group.

registered in the Brazilian Clinical Trials Registry (ReBEC RBR-2GH23P). Participation was voluntary, and all participants signed an informed consent document before taking part in the study, which was conducted in accordance with the Declaration of Helsinki with the recent amendment of Fortaleza (2013). All the individuals selected for the study were considered able to perform the exercise based on Par-Q (Canada's physical activity guide to healthy active living) [23]. Although experienced participants have had more than 18 months of experience, they were classified as recreational as they have never participated in an official competition (except those organized by the gym itself). Participants commonly perform three to five whole-body exercise sessions per week. To sample characterization, motor performance tests (deadlift and yo-yo test) and body composition evaluation were performed prior to the intervention. These evaluations are shown in the results section (Table 1).

## Intervention

On visit 4, after a 48-h rest interval from visit 3, volunteers reported to the gym in the morning. The HIFT session performed was the gymnastics WOD called the 'Cindy' workout [16, 24]. The session started with a warm-up consisting of 5 min of low-intensity running and 5 min of joint mobility and dynamic stretching exercises. This WOD consisted of as many rounds as possible of 5 pull-ups, 10 push-ups, and 15 air squats in 20 min. The assistants of the authors were in charge of counting the rounds. The techniques used for each exercise have been described in detail in the literature [24]. When necessary, small modifications of the exercises (e.g. push-up on knees and ring rows for some women and men with insufficient strength) were made [25]. The 'Cindy' workout was chosen because it caused the greatest metabolic and cardiovascular stress when compared to other protocols [14, 15, 26]. Before the start of the study, all volunteers had a familiarization session (visit 1), providing them with technical lessons on how to do the exercises. Ambient conditions for session were as follows: temperature of 25–29°C and relative humidity of 55–65%.

During the 'Cindy' workout, the subjects were monitored using an HR monitor (Polar Team Pro, Kempele, Finland). HR data were stored and subsequently extracted using the Polar Team 2 Pro system. Ratings of perceived exertion (REP), whose use was strongly recommended in a HIFT metabolic training session [6, 7, 27] were obtained using the CR10 Borg

RPE scale [28]. The subjects were asked this question: 'how hard do you feel the exercise was?'. RPE measurements were taken 30 min after the 'Cindy' workout. One hour before the training session, the volunteers consumed a standardized breakfast (approximately 320–350 cal) with a protein to fat to carbohydrate ratio of 20-35-45 (protein, fat, carbohydrate as percentage). These percentages culminated in the intake of approximately 40g carbohydrates, 17.5g protein, and 13g fat.

## Blood collection

Before (pre-ex), immediately after (post-ex), 30 min afer (post-30 min), and 24 h after (post-24 h) exercise, blood samples (~8 mL) were obtained from the antecubital vein and transferred to vacuum tubes. The blood was analyzed for leucocyte (WBC), neutrophil, lymphocyte, mono-cyte (tubes mixed with ethylenediamine tetraacetic acid (EDTA)), and CK concentrations at pre-ex, post-ex, post-30 min, and post-24 h. Cortisol (tubes with serum gel) and blood lactate concentrations (tubes with sodium fluoride) were obtained at pre-ex, post-ex, and post-30 min (Fig 1). Leucocyte, neutrophil, lymphocyte, and monocyte were analyzed using an automatic haematology analyzer (Cell-Dyn Ruby System; Abbott Laboratories, Abbott Park, IL, USA). CK and lactate were measured with commercial kits, according to the manufacturer's recom-mendations and using an automatic biochemistry analyzer (CMD 800i, Wiener Lab., Rosario, Argentina). Cortisol was measured with a commercial kit, according to the manufacturer's rec-ommendations and using an immunoassay analyzer (Abbott ARCHITECT i1000SR, Abbott Park, IL, USA).

## Statistical analyses

The normality and homogeneity of the variances were verified using the Shapiro–Wilk and Levene tests, respectively. Data are presented as mean and standard error of the estimate of the mean. In order to compare the mean values of the descriptive variables between the groups (EXP vs. NOV), an independent and a paired t-test were used. A 2 × 4 repeated measures anal-ysis of variance (ANOVA) (interaction groups [EXP vs. NOV] × time [moments]) was used to compare the blood analysis. Post hoc comparisons were performed using the Bonferroni test (with correction). Assumptions of sphericity were evaluated using the Mauchly's test. Where sphericity was violated (p<0.05), the Greenhouse–Geisser correction factor was applied. The Spearman product moment correlation was used to examine the relationship between rounds and immune system, and hormonal parameters. The magnitude of the correlation coefficients was considered weak (0.1<r< 0.3), moderate (0.4<r<0.6), and strong (r>0.7). Nonparametric statistics were used for the following variables: age, time, experience, and lymphocyte level at

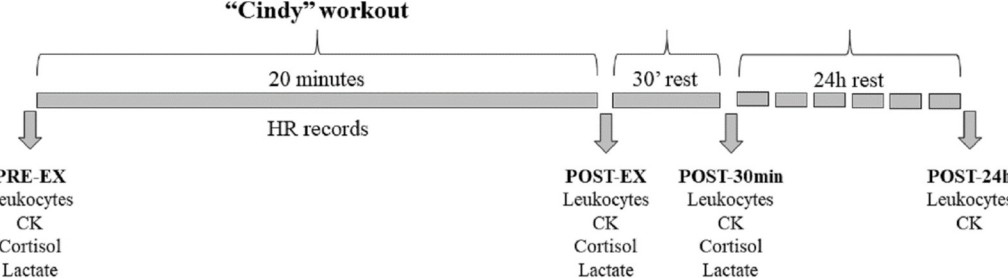

**Fig 1. Time points of blood collection.** PRE-EX = before exercise; POST-EX = immediately after exercise; POST-30min = post 30 minutes after exercise; POST-24h = 24h after exercise. HR = heart rate; CK = creatine kinase concentrations.

**Table 2. Physiological and perceptual responses under the "Cindy" workout (M ± SEM [SD]).**

| Variables | EXP (13) | NOV (10) | p-value | All (23) |
|---|---|---|---|---|
| Number of rounds | 15.1 ± 1.0 [3.5] | 10.9 ± 0.8 [2.4] | 0.004 | 13.3 ± 0.8 [3.7] |
| $HR_{av}$ (bpm) | 173.4 ± 2.6 [9.5] | 170.1 ± 2.3 [7.3] | 0.377 | 172.0 ± 1.8 [8.6] |
| $\%HR_{max}$ | 93.1 ± 0.6 [2.2] | 93.0 ± 0.8 [2.4] | 0.855 | 93.1 ± 0.5 [2.2] |
| RPE (post-30min) | 7.5 ± 0.3 [1.2] | 8.1 ± 0.3 [1.1] | 0.262 | 7.8 ± 0.2 [1.2] |
| RPE/Number of rounds ratio | 0.52 ± 0.3 [0.13] | 0.77 ± 0.6 [0.21] | 0.002 | 0.63 ± 0.4 [0.21] |

M = mean; SEM = standard error of the mean; SD = standard deviation.

$HR_{av}$ = average heart rate; $\%HR_{max}$ = percentage of maximum heart rate; RPE = rate of perceived exertion; EXP = experienced group; NOV = novice group.

pre-ex, at post-30 min, and at post-24 h, and CK activity at post-24 h. All analyses were conducted using SPSS-22.0 software (IBM, SPSS Inc., Chicago, IL, USA). Significance was set at $p < 0.05$.

## Results

Out of 224 volunteers, 118 met the inclusion criteria (minimum of 3 months of experience and the ability to perform the 'Cindy' workout). The reasons for the other volunteers being excluded from the study were as follows: 82 for consuming supplements described in the exclusion criteria; three for having 9–17 months of experience; three for having metabolic diseases; three for not completing the workout; two for having a cold; and two for not consuming the standardized breakfast. After the selection process, 23 participants completed the study.

Neither groups (EXP or NOV) differed in terms of age, sex, height, weight, deadlift 1RM test, and $HR_{max}$ in yo-yo IR1; nevertheless, BMI and body fat were lower, and $VO_{2max}$ and time experience were higher in the EXP group (Table 1).

Table 2 presents the mean number of rounds, cardiovascular variables, rate of perceived exertion, and RPE-number of rounds ratio for all participants and both groups. The mean number of rounds was higher, and RPE-number of rounds ratio was lower in the EXP in comparision with the NOV group, but $HR_{max}$, $\%HR_{max}$, and RPE at post-30 min did not reveal any significant differences between the groups.

### Leucocyte count

WBC level increased significantly following the 'Cindy' workout immediately post-ex compared to baseline values for all participants, returning to PRE values within post-30 min ($p < 0.01$). There were no differences between the groups in WBC count (Fig 2).

There was a significant increase in neutrophil count post-ex and at post-30 min, returning to pre-ex values at 24 h after the end of the protocol for all participants ($p < 0.01$). There were no differences between the groups for neutrophil count; however, only for the NOV group, post-30 min values were significantly decreased ($p = 0.020$) when compared to the values immediately post-ex (Fig 2).

Lymphocyte levels increased significantly post-ex and decreased significantly at post-30 min, returning to pre-ex values 24 h after the end of the protocol for all participants ($p < 0.01$). Lymphocyte levels in the EXP were significantly higher ($p = 0.005$) post-ex than those in NOV subjects (Fig 2).

There was a significant increase in monocyte levels post-ex, which returned to pre-ex values within post-30 min and increased again 24 h after the end the protocol for all participants ($p < 0.01$). Monocyte levels did not differ between the groups (Fig 2).

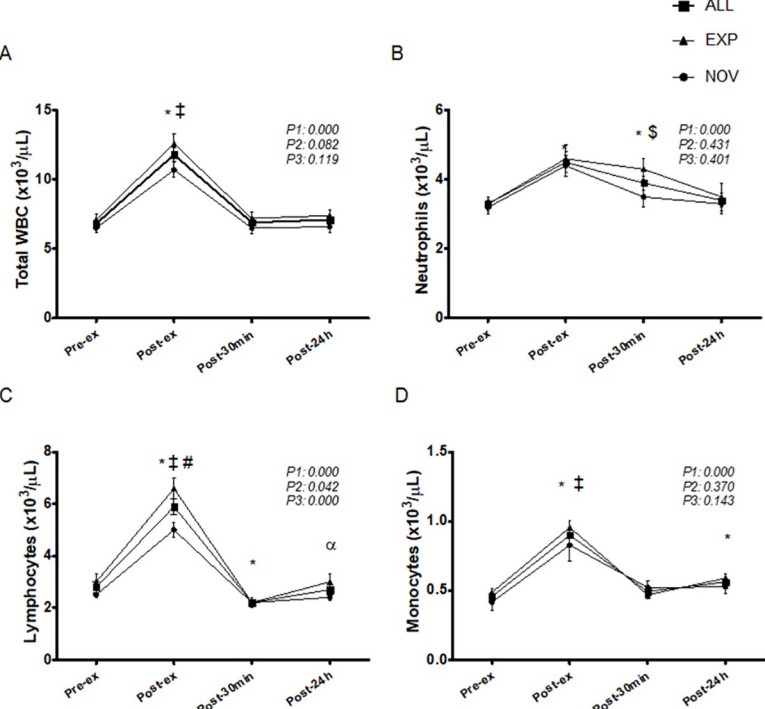

**Fig 2.** Timeline of leucocytes (A), neutrophils (B), lymphocytes (C), and monocytes (D), corresponding to before (pre-ex), immediately after (post-ex), 30 minutes after (post-30min), and 24h after (post-24h) exercise. WBC = white blood cell; EXP = experienced group; NOV = novice group. *Significantly different from pre-ex ($p < 0.05$); ‡Significantly greater than post-30min and post-24h ($p < 0.05$); ≠Significantly different from NOV group ($p < 0.05$); $Significantly lower than post-ex only for NOV ($p < 0.05$); αSignificantly greater than post-30min ($p < 0.05$). P1 = time effect; P2 = group effect; P3 = interaction group x time.

Table 3 presents the neutrophil-lymphocyte ratio (NLR) following a single 'Cindy' workout session. The NLR decreased significantly post-ex compared to baseline values, rising to values above those pre-ex and post-ex (p<0.01), and returning to pre-ex values at 24 h after the end of the protocol for all participants. There were no differences between the groups regarding the NLR. Neutrophil-cortisol ratio (NCR) increased significantly post-ex and decreased within post-30 min for all participants (p<0.01). NCR in the NOV was significantly higher post-ex (p = 0.021) than it was in the EXP participants.

Only leucocytes (r = 0.43; p = 0.040) and lymphocytes (r = 0.55; p = 0.006) post-ex showed moderate correlations with the number of rounds.

## Muscle damage

Compared to pre-ex values, there were significant increases in CK at all time points for all participants. There were no differences between immediately after, 30 min after, and 24 h after exercise time points (p>0.01). There were no differences between the groups for CK (Fig 3).

## Stress markers

As shown in Fig 4, lactate increased significantly post-ex and decreased within post-30 min, but values remained above baseline for all participants (p<0.01). There were no differences between the EXP and NOV individuals for this stress marker.

**Table 3. Pre- to post-changes in Neutrophil-Lymphocyte ratio and Neutrophil-Cortisol ratio following a single 'Cindy' workout session (M ± SEM [SD]).**

| Variables | Group | Pre-ex | Post-ex | Post-30 min | Post-24 h |
|---|---|---|---|---|---|
| **NLR** | EXP | 1.20 ± 0.14 [0.50] | 0.72 ± 0.07 [0.25] *‡ | 2.04 ± 0.22 [0.78]* | 1.28 ± 0.18 [0.65] |
| | NOV | 1.34 ± 0.10 [0.33] | 0.91 ± 0.07 [0.22] *‡ | 1.58 ± 0.12 [0.39] | 1.46 ± 0.16 [0.52] |
| | All | 1.26 ± 0.09 [0.43] | 0.80 ± 0.05 [0.25]*‡ | 1.84 ± 0.14 [0.67]& | 1.36 ± 0.12 [0.59] |
| **NCR** | EXP | 0.22 ± 0.02 [0.06] | 0.24 ± 0.02 [0.08]≠ | 0.22 ± 0.02 [0.07] | - |
| | NOV | 0.25 ± 0.02 [0.07] | 0.34 ± 0.03 [0.11]$ | 0.24 ± 0.03 [0.09] | - |
| | All | 0.23 ± 0.01 [0.07] | 0.28 ± 0.02 [0.10]*α | 0.23 ± 0.02 [0.08] | - |

M = mean; SEM = standard error of the mean; SD = standard deviation.

NLR = Neutrophil/Lymphocyte ratio; NCR: Neutrophil/Cortisol ratio; Pre-ex = before exercise; Post-ex = immediately after exercise; Post-30min = 30 minutes after exercise; Post-24h = 24 hours after exercise; EXP = experienced group; NOV = novice group.

*Significantly different from pre-ex ($p<0.01$)

‡Significantly different from post-30min and post-24h ($p<0.01$)

&Significantly different from pre-ex, post-ex, and post-24h ($p<0.01$)

αSignificantly different from post-30min ($p<0.01$).

≠Significantly different from NOV ($p<0.01$)

$Significantly different from pre-ex and post-30min only for NOV ($p<0.01$)

Neutrophil/Lymphocyte ratio = Time effect: 0.000; Group effect: 0.958; Interaction group x time: 0.371.

Neutrophil/Cortisol ratio = Time effect: 0.557; Group effect: 0.001; Interaction group x time: 0.127.

There were significant increases in cortisol levels post-ex and at post-30 min for all participants ($p<0.01$). Cortisol concentration in the NOV participants was significantly higher ($p = 0.045$) than pre-ex values only at post-30 min. The cortisol concentration in EXP subjects was significantly higher both post-ex ($p = 0.001$) and at post-30 min ($p = 0.036$) than it was in

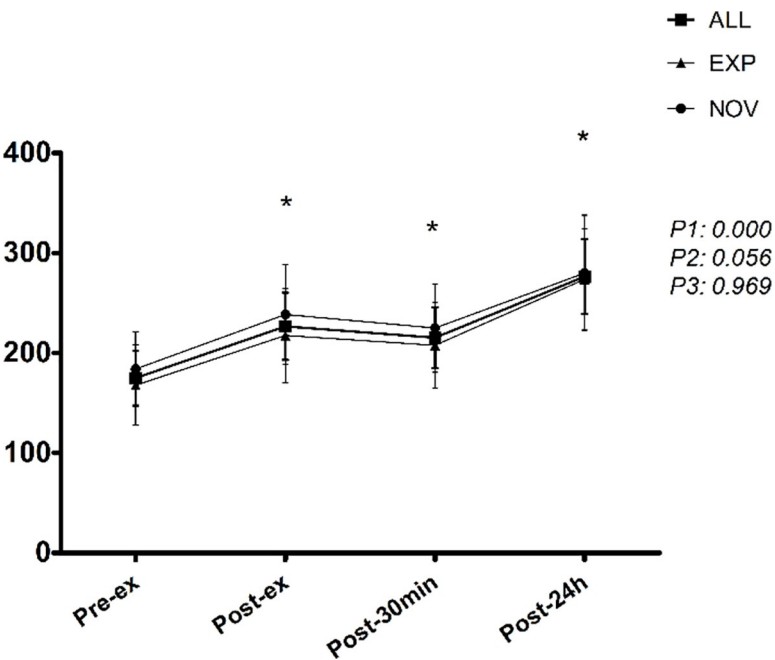

**Fig 3. Timeline of creatine kinase corresponding to before (pre-ex), immediately after (post-ex), 30 minutes after (post-30min), and 24h after (post-24h) exercise.** EXP = experienced group; NOV = novice group. *Significantly different from pre-ex ($p < 0.05$). P1 = time effect; P2 = group effect; P3 = interaction group x time.

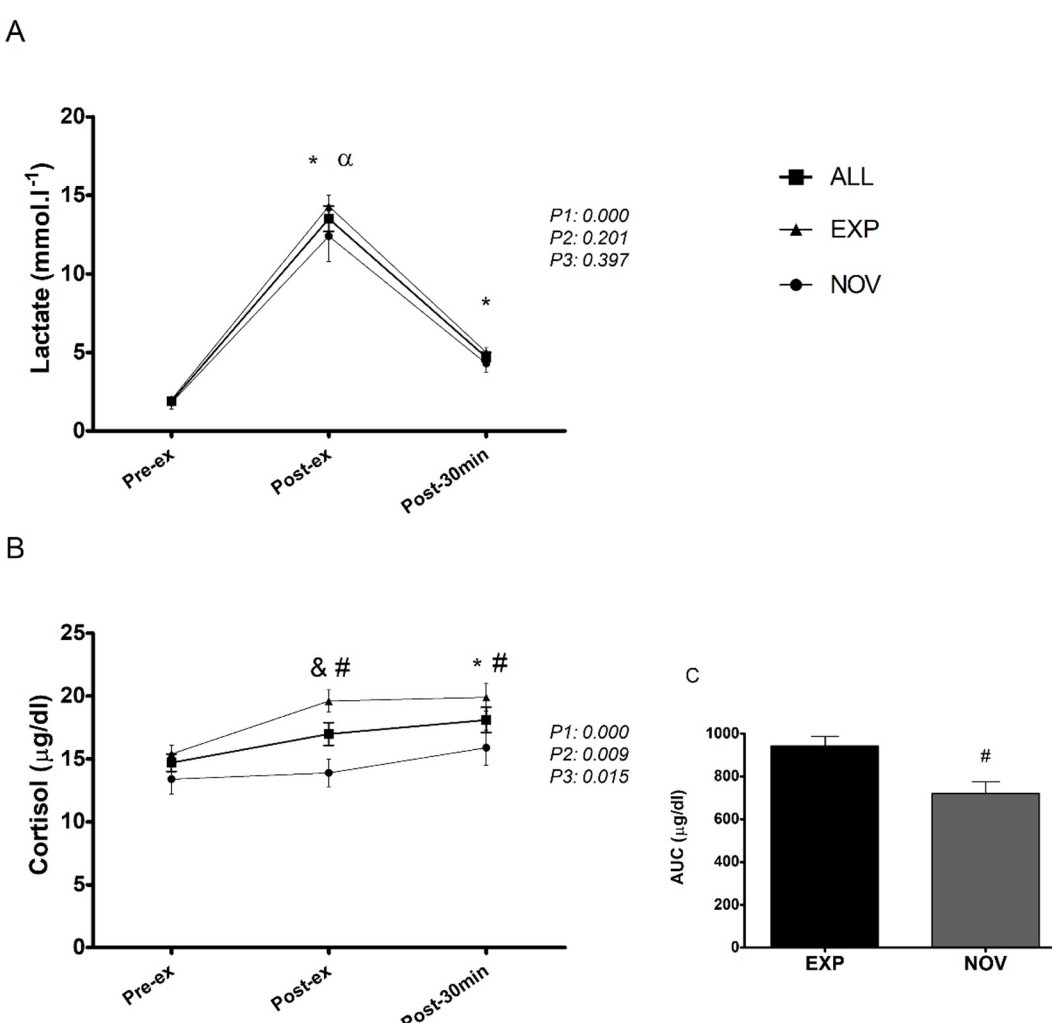

**Fig 4.** Timeline of lactate (A), cortisol (B) and area under the curve (AUC) of cortisol (C), corresponding to before (pre-ex), immediately after (post-ex) and post 30 minutes (post-30min) exercise. EXP = experienced group; NOV = novice group. *Significantly different from pre-ex ($p < 0.05$); &Significantly greater than pre-ex only for ALL and EXP ($p < 0.05$); #Significantly different between EXP and NOV group ($p < 0.05$); αSignificantly greater than post-30min ($p < 0.05$). P1 = time effect; P2 = group effect; P3 = interaction group x time.

the NOV ones. The area under the curve (AUC) of cortisol was also significantly higher in the EXP (p = 0.004) than in the NOV participants (Fig 4). Cortisol post-ex showed moderate correlations (r = 0.53; p = 0.010) with the numbers of rounds.

## Dietary analysis

Table 4 shows the daily intake of energy and macronutrients during 3 non-consecutive days. Neither groups (EXP or NOV) presented any significant differences.

## Discussion

The present study investigated whether WBC count, plasma CK activity, cortisol concentration, and lactate response are affected by an acute HIFT session. The main findings of this study were: a) after a single 'Cindy' workout session, alterations in the number of circulating

Table 4. Daily intake of energy and macronutrients during 3 non-consecutive days (M ± SEM [SD]).

| Variables | EXP (9) | NOV (3) | p-value | All (12) |
|---|---|---|---|---|
| Energy (Kcal) | 1744.0 ± 282.7 [848.0] | 1175.5 ± 321.4 [556.7] | 0.311 | 1601.9 ± 231.9 [803.4] |
| Energy/Kg (Kcal) | 23.5 ± 3.8 [11.3] | 15.5 ± 3.6 [6.2] | 0.276 | 21.5 ± 3.1 [10.6] |
| CH (g) | 203.6 ± 43.1 [129.2] | 133.4 ± 52.9 [91.6] | 0.410 | 186.1 ± 35.0 [121.1] |
| CH/Kg BM | 3.0 ± 0.5 [1.6] | 1.7 ± 0.6 [1.1] | 0.253 | 2.7 ± 0.5 [1.6] |
| PTN (g) | 93.2 ± 14.4 [43.1] | 53.7 ± 14.4 [25.0] | 0.171 | 83.3 ± 12.2 [42.3] |
| PTN/Kg BM | 1.4 ± 0.2 [0.5] | 0.7 ± 0.2 [0.3] | 0.060 | 1.2 ± 0.2 [0.5] |
| Fat (g) | 61.1 ± 6.5 [19.4] | 49.2 ± 9.6 [16.6] | 0.367 | 58.1 ± 5.4 [18.8] |

M = mean; SEM = standard error of the mean; SD = standard deviation.

BM: Body mass; CH: Carbohydrate; PTN: Protein. EXP = experienced group; NOV = novice group.

leucocytes and subsets occurred, varying slightly according to the level of training experience; b) the workout session also elicited significant increases in muscle damage; c) cortisol and lactate levels increased after the training session. To the best of our knowledge, this is the first study to analyze the acute effects of a HIFT session–gymnastics modality–on immune parameters and hormonal concentrations, and to provide fundamental information on these biomarkers. The results can be useful because of the increased number of HIFT practitioners, who may experience physiological changes in the quality and number of WBCs, thus affecting their capacity to resist common infections [29].

Unlike other studies, the participants in the present study have never engaged in competitions [5] and showed a lower level of physical fitness [15, 26], as observed by the minimum number of rounds (6 to 10 more rounds) and $VO_2max$. However, in these studies, only men and younger adults were included as participants.

Assessing the average HR achieved during the 'Cindy' workout, we obtained a mean value of 172.0 ± 8.6 bpm (93.1 ± 2.3%$HR_{max}$). According to the American College of Sports Medicine (ACSM) guidelines, this %$HR_{max}$ could be described as being close to that corresponding to maximum-intensity exercise [30]. Studies that used the 'Cindy' workout in their intervention also found high cardiovascular demand [15, 21, 31]. Therefore, we can consider that our initial intervention proposal through a HIFT session was achieved. The EXP and NOV groups had similar relative cardiovascular intensities, with 93.1 and 93.0%$HR_{max}$, respectively.

One of the findings of this study was the high work intensity reflected by blood lactate levels. The results reported in the present study are in agreement with other investigations that observed significantly elevated levels of lactate for the 'Cindy' workout in which this variable always exceeded 10 mmol/$L^{-1}$ [14, 15, 26]. In our results, even after 30 min recovery, blood lactate levels remained above baseline (>2.8 mmol/$L^{-1}$). There were no differences in lactate concentrations between the groups at any time point. In the same way, another intensity variable, such as RPE, had similar responses. RPE showed a moderate correlation with blood lactate 30 min after the session in a recent publication [7]. Comparable to lactate concentrations, there were no differences between the groups for RPE. On the other hand, when analyzing the RPE/Number of rounds ratio, we found disparities between the groups, showing that the EXP subjects perceive less effort per round.

It is believed that, even with a larger number of rounds (20 pull-ups, 40 push-ups, and 60 more air squats for the EXP group), the EXP participants showed better recovery capacity between rounds. The EXP group showed superior aerobic capacity (>1.5 mL/kg/min) than the NOV group, a fact also observed in a recently published study, in which superiority in aerobic capacity and HIFT performance was demonstrated in participants with more experience

[32]. Higher aerobic capacity has been related to high lactate removal rate, since aerobic training can promote higher synthesis of monocarboxylate transporters [33].

However, despite this evidence, it is not possible to state that the largest $VO_{2max}$ of the EXP group is responsible for the difference in the number of rounds. Previous study evaluated practitioners who competed in the 2014 CrossFit Open and/or Regional competitions and concluded that, in this population studied, which performed an average of 23.3 rounds in 'Cindy' workout, $VO_{2max}$ cannot predict performance in the CrossFit® benchmark WOD [34]. It is worth mentioning that maximum strength level revealed no differences between groups. Although the NOV group had only 3 to 8 months of experience with HIFT, they were all well versed in strength training. Therefore, it is possible to speculate that the difference between EXP and NOV is explained by local muscle endurance, a parameter not evaluated in the present study.

In the present investigation, the cortisol concentrations increased immediately after exercise and remained elevated at post-30 min. Following extremely stressful exercise, the release of hormones, including testosterone, oestrogen, and cortisol is likely to occur [35, 36], and the highest cortisol increases immediately after exercise have been observed in high-intensity exercise protocols [17, 37]. When comaparing the groups, the EXP one showed a cortisol concentration significantly higher than that of the NOV group immediately after and 30 min after exercise, including when analyzed for the AUC. This may once again be explained by the fact that the higher number of rounds performed by the EXP group might simply be a reflection of the level of training presented by the NOV group. It is worth noting that a moderate correlation between cortisol and the number of rounds at the post-ex was found. This excess circulating cortisol in the body following high-intensity exercise protocols (e.g. the 'Cindy' workout) warns of some negative health outcomes, such as the saturation of many target receptors in the repair process and in the immune response [37].

After the 'Cindy' workout, there were alterations in the number of circulating leucocytes and subsets, which varied slightly according to the level of training experience. The total number of leucocytes circulating in peripheral blood is strongly influenced by physical exercise [29, 35]. In the present study, we found a significant increase in leucocyte counts post-ex. It has been attributed to the sympathetic activation and immediate release of catecholamines during exercise, promoting the demargination of leucocytes adhered to the marginal pool, as well as cellular recruitment from storage and synthesis tissues for circulation by mechanical action [35]. In the group comparison, the EXP group showed significantly higher lymphocyte counts post-ex than did the NOV one. Additionally, the higher lymphocyte counts may be related to the ability these participants have to recover more quickly for new sessions, as their immune system has a better memory of this type of stress [38, 39]. It is important to note that moderate correlations between the number of rounds with leukocytes and lymphocytes in the post-ex were also found.

Although WBC returned to pre-ex values within post-30 min, neutrophils remained above and lymphocytes below baseline values. We believe that the neutrophils and lymphocytes results at post-30 min are a consequence of the late action of cortisol, possibly mediated by IL-6. In maximum-intensity exercise, there is an increase in several cytokines in the blood circulation, IL-6 being the main one. Generally, IL-6 comes from the skeletal muscle and from the adipose tissue. When muscle damage occurs, immune cells infiltrate in the skeletal muscle and secrete IL-6 as well, causing an elevation of weak magnitude in the plasma concentration, but just in the period necessary to repair the damaged tissue. Significant increase in IL-6 in maximal exercise is mainly due to greater release by skeletal muscle [40]. In addition, chronic low carbohydrate intake and low muscle glycogen stores can stimulate IL-6 synthesis in exercise. IL-6 elevation in exercise culminates in a series of metabolic events, such as hepatic

glycogenolysis and gluconeogenesis, fat degradation in adipocytes, muscle protein and amino acid degradation, and increased release of cortisol by the adrenal glands [41]. Increased cortisol has late effects, such as inhibition of mitogenesis and/or acceleration of lymphocyte apoptosis resulting in lymphocytopoenia, besides raising the mobilization of bone marrow neutrophils to blood, which increases the count of these cells at the end of the exercise sessions [35]. After maximum workout, lymphocytopoenia seems to have resulted from either death via apoptosis [42] or immune cell migration [43], or it may have been triggered by both [44].

In the present study, we believe that lymphocytopenia and the maintenance of high levels of neutrophils observed in the EXP group at post-30 min are a consequence of the late action of cortisol, probably in conjunction with IL-6, even though cytokine/myokine was not analyzed in the present study. Considering the NLR normality values (0.78–3.58) previously proposed [45], in the present study, although the NLR was within the normal range at all time points for both groups, at post-30 min it was observed to be higher than pre-ex only for the EXP group (more exposed to cortisol), corroborating the hypothesis that cortisol promotes a pro-neutrophil, anti-lymphocyte environment. Moreover, there was no difference between the NCR pre-ex and at post-30 min, which suggests the possible late action of cortisol on these white cells. It is worth mentioning that NCR at the post-ex does not justify the increase in neutrophils observed in the study, as expected, because in post-ex the catecholamines effects are usually more relevant.

The monocyte numbers quickly reverted to their original pre-ex values at post-30 min, and they showed delayed monocytosis at 24 h post-ex, as predicted. There is some evidence of increased monocyte count following 1.5–2.0 h of recovery after single bouts of endurance-based exercise [29, 35]. It is important to notice that none of the participants presented values above the reference for leucocyte indices, which is fundamental for optimal physical performance.

Plasma CK activity was used to assess muscle damage. CK is an intramuscular protein usually impermeable to the membrane which is unable to pass through it in its entirety. When damage occurs, there is fragility and rupture of the plasma membrane, and CK is released in the circulation and cleared from the blood by the reticuloendothelial system [17], which justifies the delayed release of CK into circulation.

A recent study evaluated the CK concentrations of 12 experienced male HIFT practitioners in two different types of WODs [46]. There was a significant increase immediately after exercise (WOD1 492 and WOD2 689 U/L$^{-1}$), and the CK peak was observed 24 h after the intervention (WOD1 673 and WOD2 864 U/L$^{-1}$), just as it occured in the present study. The initial CK level was 406 U/L$^{-1}$ in WOD1 and 566 U/L$^{-1}$ in WOD2. Another study followed nine participants for three days of HIFT competition [47]. The results showed that CK activity increased by 50% 24 h after the competition (472 U/L$^{-1}$ at baseline and 698 U/L$^{-1}$ at post-24 h). It is noteworthy that in the investigation of Tibana et al. (2019), the participants were exposed to exhaustive exercises for 3 consecutive days, whereas in our study all participants were exposed to 20 min of intervention. Additionally, the CK values presented by the referred study at pre-competition were higher than normal, demonstrating that, in this study, the participants had probably started the experiment without full recovery, similar to the results found by Timón et al. (2019).

With regard to the initial level (pre-ex) of CK, it is important to mention that, in our study, the participants had two days of absolute rest before the 'Cindy' workout, and it is probably for that reason that the initial levels of CK found in the other two studies [46, 47] were much higher than those in our findings (174 U/L-1 pre-ex). As for post-ex (immediately after), in the study by Timón et al. (2019) there was a significant CK increase when compared to pre-ex, just like observed in our study. All the studies [46, 47]–ours included–had significant CK

elevations at post-24h compared to pre-ex, as expected. With regard to the increase in CK between post-ex and post-24h, it is possible to compare our study with that of Timón et al. (2019), which adopted percentage difference (Δ) values to analyze these time points. For that, we calculated Δ values of our study and found an increase of 22%, while Timón et al. (2019), in WOD2, observed an increase of 37% in the same interval time. The reason for our percentage difference not to be such high in CK between post-ex and post-24h, when compared to study mentioned above, may be the training pattern rather than its intensity [48], as simple and more commonly performed movements were adopted in the present study (pull-ups, push-ups and air squats), while more complex moviments (WOD 2: wall ball and power clean) were performed in the other study. Comparing the absolute values observed in the other studies [46, 47] with those obtained in the present study becomes impracticable, as the initial levels were already very different.

Special concerns, such as rhabdomyolysis, must be taken into consideration, as high-intensity, short-rest protocols could induce excessive muscle damage in athletes [37]. Rhabdomyolysis is a condition in which the excessive amount of damaged muscle tissue breaks down its intracellular contents, and abnormal CK (10.000–20.000 U/L) and myoglobin levels in relation to those expected for responses to exercise are released into the circulation, leading to secondary clinical and biochemical complications [49, 50]. Such condition did not occur in our study, considering that the highest average CK concentration at post-24 h was 276 U/L, with one subject with a 765 U/L maximum value of CK.

The EXP group performed significantly more work, leading to greater potential for structural damage; however, no differences in CK concentration between the groups were found at any time point. The similarity in CK increase in both groups, even with the EXP having done more work, can be explained by the fact that this phenomenon does not depend on the amount of work itself, but on the amount of work performed in comparison with the amount of work performed previously in preceding trainings. EXP practitioners performed more work because they were used to performing more work. Hence, the proportion between work performed in the present analysis and the one usually performed prior to it was possibly similar between the EXP and NOV groups. Another likely explanation is the major differences in CK response among our volunteers, just as observed in a recent investigation [47].

Nutrient intake has a relevant impact on the parameters analyzed in the present study. There were no differences between the groups with regard to energy and macronutrient intake; however, it is important to highlight that the absence of differences may also have occurred due to the reduced number of participants in the NOV group. The dietary pattern of the participants in the present study revealed insufficient intake of energy (<50 kcal/kg/day [22]) and carbohydrates (<5g/kg/day [22]). The energy and macronutrient intake found in the present study was lower than that observed in Overtraining Syndrome-affected HIFT athletes assessed in a recently published study [9]. It is possible that the participants of the present study have lower values than those of Cadegiani et al. (2019) due to ergogenic supplement intake (e.g. creatine), thus being considered an exclusion criterion. In general, creatine consumers also ingest hypercaloric supplements or proteins because such products claim to lead to muscle hypertrophy [22]. Low carbohydrate intake can promote immunological disorders and a higher rate of proteolysis (CK elevation), probably due to mechanisms that involve greater release of IL-6 from the skeletal muscle into the blood circulation during exercise [51]. Still, it is relevant to emphasize that at pre-ex, the volunteers consumed a standardized breakfast, with adequate amounts of carbohydrates [52], which would minimize the impact of nutrition on the findings of the present study.

In this study, some limitations should be highlighted, such as: a) the absence of inflammation marker measures which would allow us to make further inferences about the immune

response; b) absence of lactate measure at 24h-post for the evaluation of the potential differences in lactate clearance speed between the groups, as a marker of speed recovery; c) absence of speed control or duration of each round; d) use of indirect test to assess $VO_{2max}$, which may have underestimated the aerobic capacity of the participants; e) low adherence to the food record, even with a number lower than the one determined by the sample calculation to detect differences between the groups; f) focus limited on HIFT exercise that has a high metabolic demand, despite knowing that a traditional HIFT session is comprised of specific exercises for the development of strength, power, and gymnastics, which are usually performed before the WOD.

Several aspects demonstrate the need for further studies. In view of the acute effects of a HIFT session on immune response, muscle damage, and stress markers, additional research for investigating other health outcomes should be conducted with both experienced and novice practitioners. In addition, the safety of HIFT should be better understood and adapted to people with low physical fitness. During the 'Cindy' workout, one man and two women in the NOV group reported discomfort and dizziness in the final minutes of the session, not being able to complete the workout. Therefore, we believe that if the way the HIFT WODs is adjusted for performance, they could be applied to anyone, regardless of experience and conditioning level. A possible strategy in exercise prescription could be either a longer recovery time during the session or a decreased total training time, with gradual progression over the training weeks.

## Conclusion

One single HIFT session elicited significant acute perturbations in WBC count, stress markers, and muscle tissue, similarly to other high-intensity training regimens of the kind. It is noteworthy that the EXP participants showed greater lymphocyte and cortisol responses than the NOV ones did.

## Supporting information

**S1 Data. Table 5.** Pre- to post-changes in outcomes of leucocytes, neutrophils, lymphocytes, monocytes, creatine kinase, lactate, cortisol and area under the curve of cortisol following a single 'Cindy' workout session (M ± SEM [SD]) *Significantly different from pre-ex ($p < 0.05$); ‡Significantly greater than post-30min and post-24h ($p < 0.05$); ≠Significantly different between EXP and NOV group ($p < 0.05$); $Significantly lower than post-ex only for NOV ($p < 0.05$); ᵅSignificantly greater than post-30min ($p < 0.05$); &Significantly greater than pre-ex only for ALL and EXP ($p < 0.05$).
(PDF)

## Acknowledgments

The authors thank all participants for their contributions, as well as coaches André Almeida and Marcos Marcelo de Oliveira for their help in conducting the interventions. Special thanks to Djane Araújo Oliveira and her team at the university hospital of the Federal University of Sergipe.

## Author Contributions

**Conceptualization:** João Henrique Gomes, Marzo Edir Da Silva-Grigoletto, Ana Mara de Oliveira e Silva, Lucindo José Quintans-Júnior.

**Data curation:** João Henrique Gomes, Crystianne Santana Franca, Ana Mara de Oliveira e Silva.

**Formal analysis:** Danilo Rodrigues Pereira da Silva.

**Investigation:** João Henrique Gomes, Crystianne Santana Franca.

**Methodology:** João Henrique Gomes, Renata Rebello Mendes, Ana Mara de Oliveira e Silva, Lucindo José Quintans-Júnior.

**Project administration:** João Henrique Gomes, Renata Rebello Mendes, Ana Mara de Oliveira e Silva, Lucindo José Quintans-Júnior.

**Resources:** Marzo Edir Da Silva-Grigoletto, Angelo Roberto Antoniolli, Ana Mara de Oliveira e Silva, Lucindo José Quintans-Júnior.

**Supervision:** João Henrique Gomes, Renata Rebello Mendes, Crystianne Santana Franca.

**Writing – original draft:** João Henrique Gomes, Renata Rebello Mendes.

**Writing – review & editing:** João Henrique Gomes, Renata Rebello Mendes, Marzo Edir Da Silva-Grigoletto, Lucindo José Quintans-Júnior.

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
