## [Decision Letter · Decision Letter 0]

7 Aug 2020

PONE-D-20-16542

Acute leucocyte, muscle damage, and stress marker responses to functional fitness programs Immune response to functional fitness programs

PLOS ONE

Dear Dr. Gomes,

Thank you for submitting your manuscript to PLOS ONE. After careful consideration, we feel that it has merit but does not fully meet PLOS ONE’s publication criteria as it currently stands. Therefore, we invite you to submit a revised version of the manuscript that addresses the points raised during the review process by the reviewers. Please, also clarify whether, and how, all data included in the manuscript is fully available.

We look forward to receiving your revised manuscript.

Kind regards,

Pedro Tauler, Ph.D.

Academic Editor

PLOS ONE

Journal Requirements:

2.Thank you for stating the following in the Acknowledgments Section of your manuscript:

[This study was financed in part by the Coordenação de Aperfeiçoamento de Pessoal de Nível Superior – Brazil (CAPES – Finance Code 001).]

 [The funders had no role in study design, data collection and analysis, decision to publish, or preparation of the manuscript]

Reviewers' comments:

Reviewer's Responses to Questions

**Comments to the Author**

1. Is the manuscript technically sound, and do the data support the conclusions?

Reviewer #1: Yes

Reviewer #2: Partly

Reviewer #3: Yes

2. Has the statistical analysis been performed appropriately and rigorously? 

Reviewer #1: Yes

Reviewer #2: Yes

Reviewer #3: Yes

3. Have the authors made all data underlying the findings in their manuscript fully available?

Reviewer #1: No

Reviewer #2: No

Reviewer #3: Yes

4. Is the manuscript presented in an intelligible fashion and written in standard English?

Reviewer #1: Yes

Reviewer #2: No

Reviewer #3: Yes

5. Review Comments to the Author

Reviewer #1: General comments:

1. This is a simple yet well thought study. Authors found a gap on a very basic knowledge and explored it appropriately. These are those kind of basic studies that are more than necessary to better characterize the effects of this specific sport, with interesting peculiarities.

2. “Functional Fitness” has also been largely used as “High Intensity Functional Training” HIFT). Since its main representative is the branded sport “CrossFit”, it was kind of challenging to uniformize a generic name. I recommend authors to change for “HIFT” throughout the manuscript.

3. Authors should be careful when using the brand name (CrossFit). Its previous owners of the brand (the brand CrossFit has been recently sold) were threatening several research groups that attempted to describe scientific reports of CrossFit, and copiously have attempted to retract studies on “CrossFit”.

Introduction

Overall comment:

I suggest a brief explanation on why FFT/HIFT may elicit different effects than those of other sports, by comparing their characteristics, which the most fundamental ones are:

1. FFT/HIFT mixes endurance, explosion, and strength within the same sport, exercise, at the same time, unlike any other sport, and may be the only activity that requires all domains of physical conditioning.

2. There are not patterns or predictable sequence of exercises as well as the characteristics of exercise execution. This leads to higher caloric expenditure since It does not allow muscles to learn patterns and “predict” the following demand patterns. Hence, organism is constantly getting adapted. Perhaps, FFT/HIFT practitioners may develop a sort of an ability to “condition faster in conditioning processes”, i.e., their bodies develop specific conditioning abilities faster than any other sport, as they are constantly submitted to “unexpected” new physical demands. This may also explain why they have been shown to exhibit hormonal conditioning processes in a larger extent compared to other sports.

The particularities above-mentioned provide substantiation to the that FFT/HIFT may promote distinct responses when compared to other sports, and deserves specific studies.

Specific comments:

Line 33-34 – Better if reworded to “whether perturbations in immune parameters occur, even after one single bout, remains unclear”

Lines 43– I recommend using “‘a single Cindy’ workout session” instead of “‘Cindy’ workout.”, since the first expression better describes the experimentation in the present manuscript.

Lines 63-65 – Authors could also mention that HIFT regimens require both endurance and strength/resistance abilities, as well as an inherent irregularity as another hallmark characteristic of HIFT.

Line 91 – “ a single Cindy’ workout training session would promote perturbations in immune parameters” instead of “ ‘Cindy’ workout would promote perturbations in immune parameters, after a training session”.

Materials and methods

Overall comment:

I missed information regarding nutritional status. We know that pre-workout macronutrient intake (besides caffeine, creatine, and others) as well as chronic, daily caloric intake and macronutrient proportion may drive responses, even the acute ones. In case authors have not assessed this information, please openly describe in the discussion and or limitations of the study that these factors could also influence the responses, but were not obtained for the present study. In case authors did collect this data, please describe it.

Specific comments:

Lines 118-119 – Were experienced participants training for competitions? This would be interesting to be mentioned, since they change their level of effort when they are training for a competition, as part of the inherent but healthy intra- and inter-individual competitiveness of FFT/HIFT.

Line 123-125 – I find it very hard to find FFT/HIFT who do not take any type of “nutritional supplements composed of vitamins, minerals, or antioxidant

compounds” (item ‘d’ of exclusion criteria). It is important to remind that caffeine, for example, has sold evidence to enhance physical performance, and would therefore be a criteria of exclusion. However, earlier in the text authors recommend athletes to stay away from caffeine on the 48 hours before the beginning of the experiment. So we admit that some may were taking caffeine (in coffee or as caps) before, and were still included. Another examples are vitamin D and zinc, overly taken by almost all athletes (which I am particularly not against at all). Here, authors may described “supplements” more specifically (which supplements would cause exclusion?)

Lines 132-139 – The ability to perform all exercises is a premise to participate in the study, as far as I could understand. From this, I have two points:

1. Authors could make explicit the fact that participants should have been able to complete all physical executions, which makes lines 132-139 as a premise to be included; and

2. The description of any result is better suited in the first paragraphs of the results section, as well as the baseline characteristics, as shown in Table 1. I wo

Line 147a – Which was the interval between visits 3 and 4? This is important.

Line 147b – Please provide details on the morning before volunteers went to CrossFit gym. Were they recommended to have any specific sort of meal? More protein, more carbohydrate, or there were no specific recommendations? This is not such an issue in case athletes did not receive any specific nutritional recommendation, because they would intuitively follow their usual eating habits, as a “real life” study. But whether they received or not nutritional recommendations must be mentioned.

Lines 171-175 – Please describe which specific assay were employed for each parameter (which is not the information “All aforementioned variables were analysed using an automatic haematology analyser (Cell-Dyn; Abbott Laboratories, Abbott Park, IL, USA”).

Lines 182-183 – Even non-normal data was presented as mean and standard error? And why not both standard error and standard deviation (SD)?

Results

Overall comments:

1. Due to the extensive list of exclusion criteria (which I do agree with, except for a more specific list of supplements to exclude participation), prior to showing results, I would like to have a brief selection process (from 45 candidates, 4 were excluded because they were taking hormonal replacement, 3 because they were practicing irregularly…After the selection process, 23 participants completed the study.).

2. A sort of proportion between number of rounds and RPE could elicit a better picture of the differences between EXP and NOV.

3. I cannot read the exact number of biochemical parameters anywhere. They are not in the figures, tables, or text. Besides findings statistical significance and illustrating results in Figures (which gives a better understanding of the results), they must be described somewhere.

4. Neutrophil-to-lymphocyte ratio has emerged as a prognostic marker, although with distinct meanings between healthy and disease states. This is easy to calculate and would provide additional information.

5. Besides catecholamines, cortisol has been repeatedly reported to increase neutrophil mobilization, not only inhibiting lymphocytes overall activity. Then, at least part of acute neutrophil release could be secondary to cortisol increase. A “cortisol adjusted” neutrophil counting (lke neutrophil:cortisol ratio) can offer a view of cortisol-independent increase of neutrophilsm, which could be particularly useful for the 30-minute after exercise collect, since cortisol takes a little bit longer than catecholamines to induce increase in neutrophils.

6. In addition to the number of rounds, the mean total duration of the Cindy workout session should be reported, since some parameters are more strictly related to duration than intensity of exercise, and number of rounds are not necessarily linearly correlated with duration, since it also depends on the speed and duration of each round.

7. Adjustments of results for differences in body fat may provide additional information, since adipose tissue also releases IL-6 and may direct- and indirectly influence responses, while groups had significant differences between body fat. The same can be applied for the amount of muscle mass.

8. A sub-group analysis of sex-specific findings would be interesting to demonstrate trends in differences. One must consider that differences in male and female physiology encompass all organs, tissues, and metabolism. Sex-specific responses, particularly in terms of timing of response, could add useful information.

Discussion:

1. CK levels are more related to changes in training patterns (any change induces more prominent increase), rather than intensity. High-intensity regimens in those that were already used to these regimens tend to have lower increase of CK, compared to those who are experiencing high-intensity trainings for the first times. Interestingly, the fact that beginners had similar CK increase compared to experienced ones eludes to the fact that the ability to get more easily conditioned may occur early in FFT/HIFT, since beginners did not had enhanced CK increased compared to experienced ones.

2. Authors expand discussions to IL-6, dedicating a whole paragraph to this, without connecting IL-6 with the present findings of the study. Discussion should focus on the findings of the present study, and although they may mention on parameters not evaluated herein, at least they need to make a connection (what IL-6 increase, which likely happened in the athletes of the present study, has to do with the findings?)

3. Authors should consider extrapolate and make descriptive comparisons with sports of other modalities, including endurance, pure strength, and explosive (ball games, for example) sports. Are these exercise-specific responses, or responses seen in general? Is it there any peculiarity in the responses not found in other sports?

4. The sequence of the discussion is a but confusing. Authors come and go to markers (for example: lactate). Please reorder to provide a logical sequence.

Specific comments:

Lines 279-282 – Since changes in IL-6 and IL-10 have been described in the introduction, I would change the sentence for: “Although cytokines have not been analysed in the present study, it was previously demonstrated that FFT, such as a CrossFit® session, immune perturbations possibly due to changes in IL-6 and IL-10 blood levels occurred because exercise protocols caused high cardiovascular, metabolic, and hormonal demands, as observed in previous 282 studies [2,9,23,27,28]”

Lines 282-284 – This has been reported in the introduction. Also, describing results “4.1 and 14.4 pg/mL” should not be alone, but compared to pre-workout levels, and should include reference ranges. Please amend.

Lines 287-288 – In order to predict that similar responses in inflammatory markers occurred in the present protocol, similar responses between parameters measured in the present study and in the study described should be present, or a description of why similar responses would be expected, something like:

“because our protocol has similar characteristics of those studies that showed increase of IL-6 and IL-10 in response to exercise, we believe that similar responses…”

Lines 289-296 – Authors should comment on the still not fully elucidated differences between the muscle-released IL-6 (as a myokine) and adipose tissue-released IL-6 (as an adipokine), in terms of biological actions. Before that, it is essential to mention that the increase of IL-6 observed in previous studies is likely originated in the muscle tissue.

Also, the promotion of leucocyte adherence caused by IL-6 would lead to decrease in WBC blood levels, since these would be attached to endothelium. Please explore this point.

Lines 197-307 – While catecholamines actions prevail over cortisol during and right after exercise, 30 minutes after the end of the training session cortisol actions prevails, while catecholamines faster return to normal levels. Cortisol promotes a pro-neutrophil anti-lymphocyte environment, leading to increased neutrophil:lymphocyte ratio (and this is why I mentioned the importance of additing this ratio as a parameter). This well explains the findings described in lines 197-198, at a larger extent than the explanation on lines 300-302.

Lines 313-314 – There is a counterargument to the point in lines 313-314, that should be mentioned: because of the higher number of rounds, the duration of the training session of EXP athletes was likely longer than NOV. Cortisol tends to be released in a training duration-manner, which means that EXP were more exposed to cortisol compared to NOV, particularly if we calculate the area under the curve (AUC) of its release (as mentioned by authors later, in lines 371-372). Hence, one would expect that lymphocyte count would be lower, not higher, in the EXP group, due to cortisol inhibitory effects.

This should be explored in the discussion.

Line 336 – How were participants exposed to an exact 20-minute duration intervention, if the number of rounds differed between them? (at least from what I understood, CK was collected in visits 4 and 5 – and in visit 4, which was the training session, I have not seen anything specifying the training duration. Otherwise number of sessions would be compromised and limited.) Please justify.

Lines 342-343 – Release of CK and myoglobin in the circulation due to damaged muscle tissue is an physiological adaptation to exercise. Authors should specify in which point and extent (if any) this muscle breakdown becomes pathological (rhabdomuolysis). A suggestion:

“Rhabdomyolysis is a condition in which the excessive amount of damaged muscle tissue breaks down its intracellular contents, and abnormal CK and myoglobin levels in relation to those expected for responses to exercise are released into the circulation, leading to secondary clinical and biochemical complications”.

Lines 350-354 – The level of increase of CK levels does not depend on the amount of work alone, but on the amount of work performed in comparison to the previous amount of work performed in previous trainings. EXP athletes performed more work because they were used to perform more work before. Hence, the proportion between work performed in the present analysis and usually performed before were possibly similar between EXP and NOV. This is likely the most suitable explanation for the lack of differences in the amplitude of CK increase.

Lines 387-388 – That would have been interesting if authors had collected lactate 24 hours after training. This would better demonstrate potential differences in lactate clearance speed between groups, as a marker of recovery speed.

Line 409 – “Regardless of experience and conditioning level”

Conclusion

Overall comment:

The way conclusion is written, it seems that there is no novelty in the present study, since muscle damage and stress markers are expected to occur in response to any minimally intense activity. A suggestion:

“This is the first report that a single FFT session elicited significant acute perturbations in WBC counting, stress markers, and muscle tissue, in an analogue manner than other high-intensity training regimens. Noteworthy, those more experienced exhibited greater lymphocyte and cortisol responses than novice ones.”

Specific comments:

Line 412 – Remove the expression “In conclusion,” Start directly with “One single..”

Lines 413-414 – “caused muscle damage”, not “increased”, because we presume that muscle damage was absent before.

Reviewer #2: This study aimed to examine acute leucocyte, muscle damage, and stress marker responses following a single bout of CrossFit and to compare the results between novice and experienced participants. To this author made a comparison between novice (3-8 months of experience; n = 10) and experienced (≥18 months; n = 13) subjects after ‘Cindy’ workout.

On the basis of their results, the authors concluded that the ‘Cindy’ workout elicited a significant perturbation in white blood cell (WBC) count, plasma creatine kinase activity, blood cortisol level, and lactate counts with increased muscle damage and stress markers and that experienced participants showed greater responses than novice.

The article is interesting and well-structured. Nevertheless, the quality of the paper needs to be improved and some corrections and implementations should be provided.

-The paper is too long, particularly the discussion section, and must be shortened.

-What is the reason for classified the subjects in EXP only through the months of training? For example, if an individual had 2 years of experience but cannot perform on single repetition of pull-up? Please explain.

-Can the authors perform a correlation between changes in immune, hormonal, and metabolic response with volume of repetitions? This can explain the higher hormonal and immune response in EXP as compared to NOV.

-Can the authors explain why the VO2 of experienced volunteers is below those reported in previous studies? Bellar et al., Biol Sport. 2015. Tibana et al., Sports. 2019; Sousa et al., 2016 - Journal of exercise physiology online; Butcher et al., 2015 Open access journal of sports medicine

-Can the authors perform a correlation between changes in immune, hormonal, and metabolic response with physical fitness tests (cardiovascular and muscle strength)? Are the subjects with better physical conditioning less responsive to muscle perturbation?

It is important to note that a functional fitness training session does not only include metabolic conditioning (for example Cindy), but includes the development of strength, power and gymnastics.

-The author can include some recent articles to increase the quality of the discussion:

Posnakidis et al. High-Intensity Functional Training Improves Cardiorespiratory Fitness and Neuromuscular Performance Without Inflammation or Muscle Damage. 2020

Falk Neto et al. Session RPE is a superior method to monitor internal training loads of functional fitness training sessions performed at different intensities when compared to training impulse. 2020.

Mangine et al. Physiological Differences Between Advanced CrossFit Athletes, Recreational CrossFit Participants, and Physically-Active Adults. 2020.

Poderoso et al. Gender Differences in Chronic Hormonal and Immunological Responses to CrossFit. 2019

Tibana RA et al. Is Perceived Exertion a Useful Indicator of the Metabolic and Cardiovascular Responses to a Metabolic Conditioning Session of Functional Fitness? 2019.

Tibana RA et al. Lactate, Heart Rate and Rating of Perceived Exertion Responses to Shorter and Longer Duration CrossFit® Training Sessions

Reviewer #3: The manuscript is very well done. Additionally, there is little scientific literature addressing these types of stress markers, muscle damage, and white blood cells in Crossfit. For this reason, I think it has the potential to be published in one.

The introduction follows a common thread with the objectives statement.

The research design is appropriate and well described.

The results are well explained and easy to understand. And the discussion of the results addresses each of the variables analyzed.

Results

P. 10, line 228 There were no differences between immediately after, 30 min, and 24 h after exercise time points (p < 0.01). Modify by p > 0.01.

6. PLOS authors have the option to publish the peer review history of their article (what does this mean?). If published, this will include your full peer review and any attached files.

Reviewer #1: **Yes: **Flavio A. Cadegiani

Reviewer #2: **Yes: **Ramires A. Tibana

Reviewer #3: No

---

## [Author Response · Author response to Decision Letter 0]

13 Sep 2020

The responses for each reviewer comments are below as in the file "response to reviewers" 

Reviewer #1: 

General comments:

1. This is a simple yet well thought study. Authors found a gap on a very basic knowledge and explored it appropriately. These are those kind of basic studies that are more than necessary to better characterize the effects of this specific sport, with interesting peculiarities.

Answer: Dear reviewer, we are very grateful for all the notes and suggestions proposed, as they certainly contributed to the improvement of the quality of our study. We were very excited about your words.

2. “Functional Fitness” has also been largely used as “High Intensity Functional Training” HIFT). Since its main representative is the branded sport “CrossFit”, it was kind of challenging to uniformize a generic name. I recommend authors to change for “HIFT” throughout the manuscript.

Answer: We agreed and changed it according to the suggestion. Throughout the entire manuscript the term “Functional Fitness” was replaced by High-Intensity Functional Training” (HIFT). 

3. Authors should be careful when using the brand name (CrossFit). Its previous owners of the brand (the brand CrossFit has been recently sold) were threatening several research groups that attempted to describe scientific reports of CrossFit, and copiously have attempted to retract studies on “CrossFit”.

Answer: Thanks for the alert. We reduced the use of the term “CrossFit”, being included only in sections where it was considered extremely necessary.

Introduction

Overall comment:

I suggest a brief explanation on why FFT/HIFT may elicit different effects than those of other sports, by comparing their characteristics, which the most fundamental ones are:

1. FFT/HIFT mixes endurance, explosion, and strength within the same sport, exercise, at the same time, unlike any other sport, and may be the only activity that requires all domains of physical conditioning.

2. There are not patterns or predictable sequence of exercises as well as the characteristics of exercise execution. This leads to higher caloric expenditure since It does not allow muscles to learn patterns and “predict” the following demand patterns. Hence, organism is constantly getting adapted. Perhaps, FFT/HIFT practitioners may develop a sort of an ability to “condition faster in conditioning processes”, i.e., their bodies develop specific conditioning abilities faster than any other sport, as they are constantly submitted to “unexpected” new physical demands. This may also explain why they have been shown to exhibit hormonal conditioning processes in a larger extent compared to other sports.

The particularities above-mentioned provide substantiation to the that FFT/HIFT may promote distinct responses when compared to other sports, and deserves specific studies.

Answer: We agreed and changed it, according to the suggestion. We added a reference (Feito Y, Heinrich K, Butcher S, Poston W. High-Intensity Functional Training (HIFT): Definition and Research Implications for Improved Fitness. Sports. 2018;6: 76. doi:10.3390/sports6030076.) that discusses the definition of HIFT to better clarify the concepts mentioned by the reviewer. Lines 65-69

Specific comments:

Lines 33-34 – Better if reworded to “whether perturbations in immune parameters occur, even after one single bout, remains unclear”.

Answer: We agreed and changed the sentence. Lines 33-34 

Line 43– I recommend using “‘a single Cindy’ workout session” instead of “‘Cindy’ workout.”, since the first expression better describes the experimentation in the present manuscript.

Answer: We agreed and changed the words throughout the text. First replacement in Line 43.

Lines 63-65 – Authors could also mention that HIFT regimens require both endurance and strength/resistance abilities, as well as an inherent irregularity as another hallmark characteristic of HIFT.

Answer: We agreed and added a sentence considering the observation in the paragraph, referencing the due study (Cadegiani FA, Kater CE, Gazola M. Clinical and biochemical characteristics of high-intensity functional training (HIFT) and overtraining syndrome: findings from the EROS study (The EROS-HIFT). J Sports Sci. 2019;37: 1296–1307. doi:10.1080/02640414.2018.1555912.) Lines 70-72

Line 91 – “a single Cindy’ workout training session would promote perturbations in immune parameters” instead of “ ‘Cindy’ workout would promote perturbations in immune parameters, after a training session”.

Answer: We agreed and changed the words in the text. Lines 99-100

Materials and methods

Overall comment:

I missed information regarding nutritional status. We know that pre-workout macronutrient intake (besides caffeine, creatine, and others) as well as chronic, daily caloric intake and macronutrient proportion may drive responses, even the acute ones. In case authors have not assessed this information, please openly describe in the discussion and or limitations of the study that these factors could also influence the responses, but were not obtained for the present study. In case authors did collect this data, please describe it.

Answer: We agree that pre-workout macronutrient intake as well as chronic daily caloric intake and macronutrient proportion may drive responses. So, we have additional information regarding nutritional status that have been included in the revised manuscript, and we believe they can enrich the work.

Pre-workout meal: Participants were instructed on the importance of pre-workout diet, and for that reason, a standardized meal was provided for that moment. Even two participants were excluded from the study because they did not eat the standardized meal. In the manuscript originally sent, we had mentioned the characteristics of the pre-workout intake (lines 164-167 / Lines 183-186 in the revised manuscript): “One hour before the training session, the volunteers consumed a standardized breakfast (approximately 320–350 calories) with a protein to fat to carbohydrate ratio of 20:35:45 (protein, fat, carbohydrate as percentage)”. In the revised version of the manuscript, we include the following complementation: these percentages culminated in the intake of approximately 40g of carbohydrates, 17.5g of protein and 13g of fat (lines 186-187). We hope that this information is satisfactory.

Chronic energy and macronutrient intake: our initial proposal also included this type of analysis, and for that, participants were asked to record three non-consecutive days of their food intake, for further evaluation of energy intake and macronutrients using the software “Dietpro” version 5.1; however, adherence to the delivery of such forms was only 52.2%, that is, of the 23 participants, only 12 delivered the completed forms, being 9 from the group EXP and only 3 from NOV. For this reason, we had given up on including such data in the publication, however, at the suggestion of the reviewers, we included the data, and pointed out its limitations, with regard to the reduced number of participants in the beginning group. Nutritional data analysis (12 participants) was inserted in the results section (table 4).

 Due to the inclusion of the information regarding nutritional status, changes were made to the manuscript, in the following sections:

Materials and methods: Lines 117-122

Results: Table 4 has been included, as well as its description in lines 348-353. 

Discussion: Lines 520-536

Limitations: Lines 542-544

Lines 118-119 – Were experienced participants training for competitions? This would be interesting to be mentioned, since they change their level of effort when they are training for a competition, as part of the inherent but healthy intra- and inter-individual competitiveness of FFT/HIFT.

Answer: The experienced participants were not training for competitions. We added the sentence in the paragraph showing the profile of experienced participants. Lines 155-157

Line 123-125 – I find it very hard to find FFT/HIFT who do not take any type of “nutritional supplements composed of vitamins, minerals, or antioxidant compounds” (item ‘d’ of exclusion criteria). It is important to remind that caffeine, for example, has sold evidence to enhance physical performance, and would therefore be a criteria of exclusion. However, earlier in the text authors recommend athletes to stay away from caffeine on the 48 hours before the beginning of the experiment. So we admit that some may were taking caffeine (in coffee or as caps) before, and were still included. Another examples are vitamin D and zinc, overly taken by almost all athletes (which I am particularly not against at all). Here, authors may described “supplements” more specifically (which supplements would cause exclusion?). 

Answer: We appreciate your observation. We agree that there was a lack of details in the material and methods section to clarify these informations. It was really difficult to find participants who were not consuming supplements that contained: a) antioxidant compounds in last six weeks and b) ergogenic compounds in last four months; for this reason, it is important to mention that this was the criterion that most excluded volunteers from the research (82 exclusions). The description of the selection process was included at the beginning of the results section (lines 226-227).

 Supplements considered as exclusion criteria were those containing antioxidant compounds in last six weeks, as well as those considered ergogenic (ISSN, 2018), such as caffeine, creatine, beta-alanine, nitrate and bicarbonates. The consumption of any of these ergogenic supplements in the last four months would exclude the volunteer. This information was included in the revised version of the manuscript on lines 142-145.

 About caffeine, specifically, cited in line 112 of the original manuscript: participants were advised to avoid caffeine, but we were referring to caffeine-rich drinks or food (e.g., tea, coffee, and chocolate), as they are products capable of reducing body hydration. This information has been included in the Line 122. 

The term “supplement” for caffeine was not used in line 112 of the original manuscript, as this consumption was considered an exclusion criterion, the authors assumed that there would be no participant with this profile. 

Lines 132-139 – The ability to perform all exercises is a premise to participate in the study, as far as I could understand. From this, I have two points:

1. Authors could make explicit the fact that participants should have been able to complete all physical executions, which makes lines 132-139 as a premise to be included; and

Answer: We agreed and changed it according to the suggestion. We added as a premise for participation in the study the inclusion criteria. Line 136.

2. The description of any result is better suited in the first paragraphs of the results section, as well as the baseline characteristics, as shown in Table 1.

Answer: We agreed and changed it according to the suggestion. We moved the description of the baseline characteristics and “Table 1” to the results section. Lines 235-239.

Line 147a – Which was the interval between visits 3 and 4? This is important.

Answer: We agreed and added this information. Line 165

Line 147b – Please provide details on the morning before volunteers went to CrossFit gym. Were they recommended to have any specific sort of meal? More protein, more carbohydrate, or there were no specific recommendations? This is not such an issue in case athletes did not receive any specific nutritional recommendation, because they would intuitively follow their usual eating habits, as a “real life” study. But whether they received or not nutritional recommendations must be mentioned. 

Answer: We agree that pre-workout macronutrient intake may drive responses. Participants were instructed on the importance of pre-workout diet, and for that reason, a standardized meal was provided for that moment. In the original manuscript (lines 164-167), we had mentioned the characteristics of the pre-workout intake: “One hour before the training session, the volunteers consumed a standardized breakfast (approximately 320–350 calories) with a protein to fat to carbohydrate ratio of 20:35:45 (protein, fat, carbohydrate as percentage)” (Lines 183-186 in the revised manuscript). In the revised version of the manuscript, we include the following complementation: these percentages culminated in the intake of approximately 40g of carbohydrates, 17.5g of protein and 13g of fat (lines 186-187). We hope that this information is satisfactory. Lines 186-187 in the revised manuscript.

Lines 171-175 – Please describe which specific assay were employed for each parameter (which is not the information “All aforementioned variables were analysed using an automatic haematology analyser (Cell-Dyn; Abbott Laboratories, Abbott Park, IL, USA”).

Answer: We agreed and added this information, describing in detail the specific assay for these analyses. Lines 195-201.

Lines 182-183 – Even non-normal data was presented as mean and standard error? And why not both standard error and standard deviation (SD)?

Answer: Non-normal data were presented as mean to allow comparison with other studies (Mangine et al., 2020; Poderoso et al., 2019; and Tibana et al., 2019). As suggested, we have included standard deviation in all the tables. Additionally, we include a new table (table 5) in “supporting information” with SD data referring to the figures.

Results

Overall comments:

1. Due to the extensive list of exclusion criteria (which I do agree with, except for a more specific list of supplements to exclude participation), prior to showing results, I would like to have a brief selection process (from 45 candidates, 4 were excluded because they were taking hormonal replacement, 3 because they were practicing irregularly…After the selection process, 23 participants completed the study).

Answer: We agreed and added this information, describing in detail the selection process. Lines 224-230.

2. A sort of proportion between number of rounds and RPE could elicit a better picture of the differences between EXP and NOV.

Answer: We agreed and added this information in table 2 (Line 250). The inclusion of this result led to the addition of a sentence in the discussion section (Lines 388-389).

3. I cannot read the exact number of biochemical parameters anywhere. They are not in the figures, tables, or text. Besides findings statistical significance and illustrating results in Figures (which gives a better understanding of the results), they must be described somewhere.

Answer: We agreed and inserted a new table (table 5). Table 5 is available at supporting information

4. Neutrophil-to-lymphocyte ratio has emerged as a prognostic marker, although with distinct meanings between healthy and disease states. This is easy to calculate and would provide additional information. 

Answer: We agreed and added this information. It was calculated the neutrophil/lymphocyte ratio in both groups, and these results were described in table 3 and cited in lines 284-288. The inclusion of these results led to the addition of a paragraph in the discussion section (lines 450-454)

5. Besides catecholamines, cortisol has been repeatedly reported to increase neutrophil mobilization, not only inhibiting lymphocytes overall activity. Then, at least part of acute neutrophil release could be secondary to cortisol increase. A “cortisol adjusted” neutrophil counting (lke neutrophil:cortisol ratio) can offer a view of cortisol-independent increase of neutrophilsm, which could be particularly useful for the 30-minute after exercise collect, since cortisol takes a little bit longer than catecholamines to induce increase in neutrophils.

and and neutrophils-to-cortisol ratio 

Answer: We agreed and added this information. It was calculated the neutrophil/cortisol ratio in both groups, and these results were described in table 3 and cited in lines 288-290. The inclusion of these results led to the addition of a paragraph in the discussion section (line 454-458)

6. In addition to the number of rounds, the mean total duration of the Cindy workout session should be reported, since some parameters are more strictly related to duration than intensity of exercise, and number of rounds are not necessarily linearly correlated with duration, since it also depends on the speed and duration of each round.

Answer: We agreed with the observation of the esteemed reviewer. In this case, we put as a limitation of the study not having exactly controlled the real effort time of each participant (line 541)

7. Adjustments of results for differences in body fat may provide additional information, since adipose tissue also releases IL-6 and may direct- and indirectly influence responses, while groups had significant differences between body fat. The same can be applied for the amount of muscle mass.

Answer: We agreed with the observation. The Spearman product moment correlation was used, but no correlation was found between body composition and cortisol and WBC. If we had analyzed IL-6, we might have obtained a better correlation.

8. A sub-group analysis of sex-specific findings would be interesting to demonstrate trends in differences. One must consider that differences in male and female physiology encompass all organs, tissues, and metabolism. Sex-specific responses, particularly in terms of timing of response, could add useful information.

Answer: We agreed with the observation. We will soon be submitting a new manuscript comparing men and women.

Discussion:

1. CK levels are more related to changes in training patterns (any change induces more prominent increase), rather than intensity. High-intensity regimens in those that were already used to these regimens tend to have lower increase of CK, compared to those who are experiencing high-intensity trainings for the first times. Interestingly, the fact that beginners had similar CK increase compared to experienced ones eludes to the fact that the ability to get more easily conditioned may occur early in FFT/HIFT, since beginners did not had enhanced CK increased compared to experienced ones.

Answer: We agreed and appreciate so much the collaboration. We explored this point of view. (Lines 493-498)

2. Authors expand discussions to IL-6, dedicating a whole paragraph to this, without connecting IL-6 with the present findings of the study. Discussion should focus on the findings of the present study, and although they may mention on parameters not evaluated herein, at least they need to make a connection (what IL-6 increase, which likely happened in the athletes of the present study, has to do with the findings?)

Answer: We agreed with the observation. We reformulated the paragraphs referring to the discussion about IL-6, establishing a greater connection with our findings (Lines 428-444)

3. Authors should consider extrapolate and make descriptive comparisons with sports of other modalities, including endurance, pure strength, and explosive (ball games, for example) sports. Are these exercise-specific responses, or responses seen in general? Is it there any peculiarity in the responses not found in other sports?

Answer: Excellent observation. Our opinion is that it would not be appropriate to extrapolate to other sports or exercises because the protocol used in the present study is quite different. From a biomechanical point of view, no sport (individual or collective) resembles Cindy. From the metabolic and cardiovascular point of view, it is possible that some endurance sports are close to what was observed in the present study, but with different duration. In this sense, we chose not to extrapolate, agreeing with Cadegiani et al. 2019(J Sports Sci), who highlighted the characteristics of HIFT as follows: "Compared to other sport modalities, CF has unique patterns, as it requires multiple abilities, given the complexity and irregularity of its activities”. 

4. The sequence of the discussion is a but confusing. Authors come and go to markers (for example: lactate). Please reorder to provide a logical sequence.

Answer: We agreed and reordered the paragraphs, improving the fluidity of the reading.

Specific comments:

Lines 279-282 – Since changes in IL-6 and IL-10 have been described in the introduction, I would change the sentence for: “Although cytokines have not been analysed in the present study, it was previously demonstrated that FFT, such as a CrossFit® session, immune perturbations possibly due to changes in IL-6 and IL-10 blood levels occurred because exercise protocols caused high cardiovascular, metabolic, and hormonal demands, as observed in previous studies [2,9,23,27,28]”

Answer: In order to focus on discussing the study's findings, we chose to reduce the paragraphs related to IL-6 and IL-10. Thus, this paragraph above was excluded from the manuscript (Lines 428-444), since the relationship between IL-6 and our findings was described in lines 428-444.

Lines 282-284 – This has been reported in the introduction. Also, describing results “4.1 and 14.4 pg/mL” should not be alone, but compared to pre-workout levels, and should include reference ranges. Please amend.

Answer: As we did not analyze interleukins in our study, as suggested by the esteemed reviewer, we decided not to detail the concentrations of this parameter in our discussion, as we would not be able to compare them. So, the paragraph was excluded from the manuscript.

Lines 287-288 – In order to predict that similar responses in inflammatory markers occurred in the present protocol, similar responses between parameters measured in the present study and in the study described should be present, or a description of why similar responses would be expected, something like:

“because our protocol has similar characteristics of those studies that showed increase of IL-6 and IL-10 in response to exercise, we believe that similar responses…”

Answer: In order to focus on discussing the study's findings, we chose to reduce the paragraphs related to IL-6 and IL-10. Thus, this paragraph was excluded from the manuscript. 

Lines 289-296 – Authors should comment on the still not fully elucidated differences between the muscle-released IL-6 (as a myokine) and adipose tissue-released IL-6 (as an adipokine), in terms of biological actions. Before that, it is essential to mention that the increase of IL-6 observed in previous studies is likely originated in the muscle tissue.

Also, the promotion of leucocyte adherence caused by IL-6 would lead to decrease in WBC blood levels, since these would be attached to endothelium. Please explore this point. 

Answer: Some details on IL-6 synthesis were discussed in lines 428-444. 

Lines 297-307 – While catecholamines actions prevail over cortisol during and right after exercise, 30 minutes after the end of the training session cortisol actions prevails, while catecholamines faster return to normal levels. Cortisol promotes a pro-neutrophil anti-lymphocyte environment, leading to increased neutrophil:lymphocyte ratio (and this is why I mentioned the importance of additing this ratio as a parameter). This well explains the findings described in lines 197-198, at a larger extent than the explanation on lines 300-302.

Answer: We appreciate the great suggestion, and have included this important parameter in our discussion. We believe that it has added relevant value to our study (Lines 447-454).

Lines 313-314 – There is a counterargument to the point in lines 313-314, that should be mentioned: because of the higher number of rounds, the duration of the training session of EXP athletes was likely longer than NOV. Cortisol tends to be released in a training duration-manner, which means that EXP were more exposed to cortisol compared to NOV, particularly if we calculate the area under the curve (AUC) of its release (as mentioned by authors later, in lines 371-372). Hence, one would expect that lymphocyte count would be lower, not higher, in the EXP group, due to cortisol inhibitory effects.

This should be explored in the discussion.

Answer: We appreciate the great suggestion, and have included this important parameter in our results (figure 4 and Lines 332-333) and discussion (Line 408). We believe that it has added relevant value to our study. 

Line 336 – How were participants exposed to an exact 20-minute duration intervention, if the number of rounds differed between them? (at least from what I understood, CK was collected in visits 4 and 5 – and in visit 4, which was the training session, I have not seen anything specifying the training duration. Otherwise number of sessions would be compromised and limited.) Please justify.

Answer: The Cindy workout is based on Crossfit's AMRAP (As Many Rounds As Possible) concept. In this type of training, the practitioner has a predetermined duration to perform the greatest number of rounds / repetitions of a given sequence of exercises. Therefore, as much as they had a different number of rounds, the total duration of the effort was 20 minutes. We were very strict in relation to the collection control, timing exactly for each participant the exact time during the intervention. To date, we have found no studies that have monitored the real time of effort during these 20 minutes of Cindy workout. This observation we include in the limitations of the present study (Line 541).

Lines 342-343 – Release of CK and myoglobin in the circulation due to damaged muscle tissue is an physiological adaptation to exercise. Authors should specify in which point and extent (if any) this muscle breakdown becomes pathological (rhabdomuolysis). A suggestion:

“Rhabdomyolysis is a condition in which the excessive amount of damaged muscle tissue breaks down its intracellular contents, and abnormal CK and myoglobin levels in relation to those expected for responses to exercise are released into the circulation, leading to secondary clinical and biochemical complications”.

Answer: Suggestion accepted. The paragraph has been included (Lines 503-507)

Lines 350-354 – The level of increase of CK levels does not depend on the amount of work alone, but on the amount of work performed in comparison to the previous amount of work performed in previous trainings. EXP athletes performed more work because they were used to perform more work before. Hence, the proportion between work performed in the present analysis and usually performed before were possibly similar between EXP and NOV. This is likely the most suitable explanation for the lack of differences in the amplitude of CK increase.

Answer: Suggestion accepted. The paragraph has been included (Lines 511-517)

Lines 387-388 – That would have been interesting if authors had collected lactate 24 hours after training. This would better demonstrate potential differences in lactate clearance speed between groups, as a marker of recovery speed.

Answer: This is a very interesting suggestion for future studies. We inserted it as a limitation of the study in lines 539-541

Line 409 – “Regardless of experience and conditioning level”

Answer: Suggestion accepted. The sentence has been included. (Lines 555-556)

Conclusion

Overall comment:

The way conclusion is written, it seems that there is no novelty in the present study, since muscle damage and stress markers are expected to occur in response to any minimally intense activity. A suggestion:

“This is the first report that a single FFT session elicited significant acute perturbations in WBC counting, stress markers, and muscle tissue, in an analogue manner than other high-intensity training regimens. Noteworthy, those more experienced exhibited greater lymphocyte and cortisol responses than novice ones.”

Answer: We agreed and replace the paragraph. We appreciate so much for the suggestion (Lines 560-563)

Specific comments:

Line 412 – Remove the expression “In conclusion,” Start directly with “One single..” (Line 560)

Answer: We agreed and made the modification. 

Lines 413-414 – “caused muscle damage”, not “increased”, because we presume that muscle damage was absent before.

 Answer: We agreed and made the modification. 

 

Reviewer #2: This study aimed to examine acute leucocyte, muscle damage, and stress marker responses following a single bout of CrossFit and to compare the results between novice and experienced participants. To this author made a comparison between novice (3-8 months of experience; n = 10) and experienced (≥18 months; n = 13) subjects after ‘Cindy’ workout.

On the basis of their results, the authors concluded that the ‘Cindy’ workout elicited a significant perturbation in white blood cell (WBC) count, plasma creatine kinase activity, blood cortisol level, and lactate counts with increased muscle damage and stress markers and that experienced participants showed greater responses than novice.

The article is interesting and well-structured. Nevertheless, the quality of the paper needs to be improved and some corrections and implementations should be provided.

Answer: Dear reviewer, we appreciate all the suggestions. We are sure that the quality of the paper has been greatly enriched. We would like to highlight that a new revision of the English language was made in the entire manuscript.

- The paper is too long, particularly the discussion section, and must be shortened.

Answer: We agreed and reorganize the fluidity of the paragraphs in the discussion section. Certainly, the discussion became more objective, however due to the inclusion of new analyzes requested by the reviewers (RPE-Number of rounds ratio; NLR = Neutrophil / Lymphocyte ratio; NCR: Neutrophil / Cortisol ratio; area under the curve (AUC) of cortisol; Dietary analysis; and correlations) the discussion became more extensive. But we believe that it has added relevant value to our study. 

-What is the reason for classified the subjects in EXP only through the months of training? For example, if an individual had 2 years of experience but cannot perform on single repetition of pull-up? Please explain.

Answer: The question is very pertinent. For the classification of experienced and novice participants, we follow the criteria listed below: 

i) We follow the same criteria as the recent publication by Butcher et al. Relative Intensity Of Two Types Of Crossfit Exercise: Acute Circuit And High-Intensity Interval Exercise. J Fit Res. 2015., which adopted these intervals time.

ii) As for performing a single pull up, in the experienced group, all 13 performed without modifications, that is, they performed both pull up and push up without modifications. In the NOV group, of the 10 participants, 7 required modification in the pull-up or push up, or in both.

-Can the authors perform a correlation between changes in immune, hormonal, and metabolic response with volume of repetitions? This can explain the higher hormonal and immune response in EXP as compared to NOV.

Answer: We correlated the number of rounds with the hormonal parameters and the immune system. We found only the significant correlations between:

- Rounds and leukocytes in post-ex (r = 0.43; p = 0.040)

- Rounds and lymphocytes in post-ex (r = 0.55; p = 0.006)

- Rounds and cortisol in post-ex (r = 0.53; p = 0.010)

These results have been inserted in the paragraphs of the results section in lines 310-311 and 333-334. The inclusion of these results led to the addition of a sentence in the discussion section (lines 410-412 and 426-427).

-Can the authors explain why the VO2 of experienced volunteers is below those reported in previous studies? Bellar et al., Biol Sport. 2015. Tibana et al., Sports. 2019; Sousa et al., 2016, Journal of exercise physiology online; Butcher et al., 2015 Open access journal of sports medicine.

Answer: The reasons for having found VO2max values below the aforementioned studies are:

i) Characteristics of the participants in our study. Unlike the other studies, the participants in the present study were not engaged in competitions. Even the most experienced were recreational CrossFit practitioners. Although the city of Aracaju is the capital of the State of Sergipe, recreational, amateur and competitive sports receive little incentive, which makes it difficult to find highly trained or competitive people. We highlight this characteristic of our sample in the lines. Lines 155-157 and 367-368.

ii) We use the yo-yo test to predict VO2max, that is, an indirect test. According to previous publications (Stojanovic et al., 2016, Montenegrin Journal of Sports Science and Medicine; Martínez-Lagunas and Hartmann, 2014, International Journal of Sports Physiology and Performance), the yo-yo test generally underestimate VO2max. The aforementioned studies used a gas analyzer. We highlight this observation in the discussion section, as study limitations. Lines 541-542.

-Can the authors perform a correlation between changes in immune, hormonal, and metabolic response with physical fitness tests (cardiovascular and muscle strength)? Are the subjects with better physical conditioning less responsive to muscle perturbation?

Answer: We correlated physical fitness tests (cardiovascular and muscle strength) with hormonal parameters, immune systems and stress markers. We found no significant correlations in any of the combinations.

- It is important to note that a functional fitness training session does not only include metabolic conditioning (for example Cindy), but includes the development of strength, power and gymnastics.

Answer: We agreed and added the excerpt highlighting that the HIFT has, in addition to metabolic conditioning, exercises aimed at strength, power and gymnastics, as a study limitation (Lines 544-547).

-The author can include some recent articles to increase the quality of the discussion:

Answer: Dear reviewer, we appreciate so much your collaboration. We have included the excellent references suggested below.

- Posnakidis et al. High-Intensity Functional Training Improves Cardiorespiratory Fitness and Neuromuscular Performance Without Inflammation or Muscle Damage. 2020

- Falk Neto et al. Session RPE is a superior method to monitor internal training loads of functional fitness training sessions performed at different intensities when compared to training impulse. 2020. 

- Mangine et al. Physiological Differences Between Advanced CrossFit Athletes, Recreational CrossFit Participants, and Physically-Active Adults. 2020.

- Poderoso et al. Gender Differences in Chronic Hormonal and Immunological Responses to CrossFit. 2019

- Tibana RA et al. Is Perceived Exertion a Useful Indicator of the Metabolic and Cardiovascular Responses to a Metabolic Conditioning Session of Functional Fitness? 2019.

- Tibana RA et al. Lactate, Heart Rate and Rating of Perceived Exertion Responses to Shorter and Longer Duration CrossFit® Training Sessions

 

Reviewer #3: The manuscript is very well done. Additionally, there is little scientific literature addressing these types of stress markers, muscle damage, and white blood cells in Crossfit. For this reason, I think it has the potential to be published in one.

The introduction follows a common thread with the objectives statement.

The research design is appropriate and well described.

The results are well explained and easy to understand. And the discussion of the results addresses each of the variables analyzed.

Answer: Dear reviewer, we are very grateful for the compliments to our manuscript.

Results

P. 10, line 228 There were no differences between immediately after, 30 min, and 24 h after exercise time points (p < 0.01). Modify by p > 0.01.

Answer: Thank you. We change accordingly (Line 316)

---

## [Decision Letter · Decision Letter 1]

16 Nov 2020

PONE-D-20-16542R1

Acute leucocyte, muscle damage, and stress marker responses to high-intensity functional training

PLOS ONE

Dear Dr. Gomes,

Thank you for submitting your manuscript to PLOS ONE. After careful consideration, we feel that it has merit but does not fully meet PLOS ONE’s publication criteria as it currently stands. Therefore, we invite you to submit a revised version of the manuscript that addresses the point raised during the review process by reviewer 2.

We look forward to receiving your revised manuscript.

Kind regards,

Pedro Tauler, Ph.D.

Academic Editor

PLOS ONE

Reviewers' comments:

Reviewer's Responses to Questions

**Comments to the Author**

1. If the authors have adequately addressed your comments raised in a previous round of review and you feel that this manuscript is now acceptable for publication, you may indicate that here to bypass the “Comments to the Author” section, enter your conflict of interest statement in the “Confidential to Editor” section, and submit your "Accept" recommendation.

Reviewer #1: All comments have been addressed

Reviewer #2: All comments have been addressed

Reviewer #3: All comments have been addressed

2. Is the manuscript technically sound, and do the data support the conclusions?

Reviewer #1: Yes

Reviewer #2: Partly

Reviewer #3: Yes

3. Has the statistical analysis been performed appropriately and rigorously? 

Reviewer #1: Yes

Reviewer #2: Yes

Reviewer #3: Yes

4. Have the authors made all data underlying the findings in their manuscript fully available?

Reviewer #1: Yes

Reviewer #2: (No Response)

Reviewer #3: Yes

5. Is the manuscript presented in an intelligible fashion and written in standard English?

Reviewer #1: Yes

Reviewer #2: Yes

Reviewer #3: Yes

6. Review Comments to the Author

Reviewer #1: Authors have fully addressed my suggestions, and raised the quality of their manuscript. Very few studies on HIFT in the world have the quality of the present paper. Congratulations!

Reviewer #2: Congratulate the authors for the effort done in improving the paper and addressing my concerns. Nevertheless, I still think some issues need clarity.

"We believe that the difference in VO2max is what mainly caused the EXP group to perform more rounds and maintain the same alterations in lactate concentration, as the maximum strength level revealed no differences between groups. Although the NOV group had only 3 to 8 months of experience with HIFT, they were all well versed in strength training."

This paragraph is not supported by scientific evidence. Butcher et al., (2015) concluded that CrossFit benchmark Cindy performance cannot be predicted by VO2. Perhaps the difference between EXP and NOV is explained by Local Muscle Endurance.

Butcher et al. Open Access J Sports Med. 2015 Jul 31;6:241-7. d

Reviewer #3: (No Response)

7. PLOS authors have the option to publish the peer review history of their article (what does this mean?). If published, this will include your full peer review and any attached files.

Reviewer #1: **Yes: **Flavio A. Cadegiani, MD, MSc, Ph.D

Reviewer #2: **Yes: **Ramires Alsamir Tibana

Reviewer #3: No

---

## [Author Response · Author response to Decision Letter 1]

16 Nov 2020

PONE-D-20-16542R1

Acute leucocyte, muscle damage, and stress marker responses to high-intensity functional training

Reviewer #1: 

General comments:

Authors have fully addressed my suggestions, and raised the quality of their manuscript. Very few studies on HIFT in the world have the quality of the present paper. Congratulations!

Answer: Dear reviewer, we are flattered by your words. It is extremely valuable for us to receive your congratulations. Your contributions were essential for us to reach this level of quality.

Reviewer #2: 

General comments:

Congratulate the authors for the effort done in improving the paper and addressing my concerns. Nevertheless, I still think some issues need clarity.

Answer: Dear reviewer, we are very grateful for the considerations. It is a great honor for us to receive your compliments. 

Discussion

P. 16, Line paragraph - 390 "We believe that the difference in VO2max is what mainly caused the EXP group to perform more rounds and maintain the same alterations in lactate concentration, as the maximum strength level revealed no differences between groups. Although the NOV group had only 3 to 8 months of experience with HIFT, they were all well versed in strength training."

This paragraph is not supported by scientific evidence. Butcher et al., (2015) concluded that CrossFit benchmark Cindy performance cannot be predicted by VO2. Perhaps the difference between EXP and NOV is explained by Local Muscle Endurance.

Butcher et al. Open Access J Sports Med. 2015 Jul 31;6:241-7. D

Answer: We agree with the observation and thank you for the indication of the article "Do physiological measures predict selected CrossFit (®) benchmark performance?”. Therefore, we reformulated the lines mentioned according to the information described in that article.

(Line 392-406)

---

## [Decision Letter · Decision Letter 2]

19 Nov 2020

Acute leucocyte, muscle damage, and stress marker responses to high-intensity functional training

PONE-D-20-16542R2

Dear Dr. João Henrique Gomes,

We’re pleased to inform you that your manuscript has been judged scientifically suitable for publication and will be formally accepted for publication once it meets all outstanding technical requirements.

Kind regards,

Pedro Tauler, Ph.D.

Academic Editor

PLOS ONE

Additional Editor Comments (optional):

Reviewers' comments:

Reviewer's Responses to Questions

**Comments to the Author**

1. If the authors have adequately addressed your comments raised in a previous round of review and you feel that this manuscript is now acceptable for publication, you may indicate that here to bypass the “Comments to the Author” section, enter your conflict of interest statement in the “Confidential to Editor” section, and submit your "Accept" recommendation.

Reviewer #2: All comments have been addressed

2. Is the manuscript technically sound, and do the data support the conclusions?

Reviewer #2: Yes

3. Has the statistical analysis been performed appropriately and rigorously? 

Reviewer #2: Yes

4. Have the authors made all data underlying the findings in their manuscript fully available?

Reviewer #2: (No Response)

5. Is the manuscript presented in an intelligible fashion and written in standard English?

Reviewer #2: Yes

6. Review Comments to the Author

Reviewer #2: (No Response)

7. PLOS authors have the option to publish the peer review history of their article (what does this mean?). If published, this will include your full peer review and any attached files.

Reviewer #2: **Yes: **Ramires A. Tibana

---

## [Editor Report · Acceptance letter]

23 Nov 2020

PONE-D-20-16542R2 

Acute leucocyte, muscle damage, and stress marker responses to high-intensity functional training 

Dear Dr. Gomes:

I'm pleased to inform you that your manuscript has been deemed suitable for publication in PLOS ONE. Congratulations! Your manuscript is now with our production department. 

Kind regards, 

on behalf of

Dr. Pedro Tauler 

Academic Editor

PLOS ONE